# Petrogenesis, Ore Mineralogy, and Fluid Inclusion Studies of the Tagu Sn–W Deposit, Myeik, Southern Myanmar

**Kyaw Thu Htun [1,2,*], Kotaro Yonezu [1], Aung Zaw Myint [3], Thomas Tindell [1] and Koichiro Watanabe [1]**

[1] Department of Earth Resources Engineering, Kyushu University, Fukuoka 819-0395, Japan; yone@mine.kyushu-u.ac.jp (K.Y.); tindell-tom@mine.kyushu-u.ac.jp (T.T.); wat@mine.kyushu-u.ac.jp (K.W.)

[2] No.2 Mining Enterprise, Ministry of Natural Resources and Environmental Conservation, Naypyitaw 100604, Myanmar

[3] Department of Geology, University of Yangon, Yangon 11041, Myanmar; aungzawmyint28@gmail.com

* Correspondence: kyawthuhtun88@gmail.com; Tel.: +81-8087362970

**Abstract:** Most of the granite-related Sn–W deposits in Myanmar are located in the Western Granitoid Province (WGP) of Southeast Asia. The Tagu deposit in the southern part of the WGP is a granite related Sn–W deposit. The biotite granite is composed of quartz, feldspars (plagioclase, orthoclase, and microcline), and micas (muscovite and biotite) and belongs to S-type peraluminous granite. Abundances of large-ion lithophile elements (LILEs), such as Rb, K, and Pb, coupled with the deficiency of high-field-strength elements (HFSEs), such as Nb, P, and Ti, indicate that the parental magma for the Tagu granite was derived from the lower continental crust at syn-collisional setting. Mineralized veins consist of early-formed oxide ore minerals, such as cassiterite and wolframite, which were followed by the formation of sulfide minerals. Three main types of fluid inclusions were distinguished from the mineralized quartz veins hosted by granite and metasedimentary rocks: Type-A—two phases, liquid (L) + vapor (V) aqueous inclusions; Type-B—two phases, vapor (V) + liquid (L) vapor-rich inclusions; And type-C—three phases, liquid + $CO_2$-liquid + $CO_2$-vapor inclusions. Quartz in the veins hosted in granite corresponding with earlier deposition contains type-A, type-B, and type-C fluid inclusions, whereas that in the veins hosted in metasedimentary rocks corresponding with later deposition contains only type-A fluid inclusions. The homogenization temperatures of type-A inclusions range from 140 °C to 330 °C (mode at 230 °C), with corresponding salinities from 1.1 wt.% to 8.9 wt.% NaCl equivalent for quartz veins hosted in metasedimentary rocks, and from 230 °C to 370 °C (mode at 280 °C), with corresponding salinities from 2.9 wt.% to 10.6 wt.% NaCl equivalents for quartz veins hosted in granite. The homogenization temperatures of type-B vapor-rich inclusions in quartz veins in granite range from 310 °C to 390 °C (mode at 350 °C), with corresponding salinities from 6.7 wt.% to 12.2 wt.% NaCl equivalent. The homogenization temperatures of type-C $H_2O$–$CO_2$–NaCl inclusions vary from 270 °C to 405 °C (mode at 330 °C), with corresponding salinities from 1.8 wt.% to 5.6 wt.% NaCl equivalent. The original ascending ore fluid was probably $CO_2$-bearing fluid which evolved into two phase fluid by immiscibility due to pressure drop in the mineralization channels. Furthermore, the temperature and salinities of two-phase aqueous fluids were later most likely decreased by the mixing with meteoric water. The salinities of the type-B vapor-rich inclusions are higher than those of the type-C $CO_2$-rich inclusions, which may have resulted from $CO_2$ separation from the fluids. The escape of gases can lead to an increase in the salinity of the residual fluids. Therefore, the main ore-forming mechanisms of the Tagu Sn–W deposit are characterized by fluid immiscibility during an early stage, and fluid mixing with meteoric water in the late stage at a lower temperature.

**Keywords:** peraluminous; S-type granite; Sn–W mineralization; fluid inclusions; immiscibility

## 1. Introduction

Myanmar is endowed with ore deposits of tin, tungsten, copper, gold, gemstones, zinc, lead, nickel, and silver. Myanmar is one of the countries with the most diverse natural and mineral resources in Southeast Asia, largely reflecting its geological history [1]. There are at least three world class deposits, including Bawdwin (lead–zinc–silver), Monywa (copper), and Mawchi (tin–tungsten). Myanmar can be divided into three principal metallogenic provinces: The Wuntho–Popa Arc, comprising subduction-related granites associated with porphyry-type copper–gold–molybdenum and epithermal gold mineralization; the Mogok–Mandalay–Mergui Belt hosting both significant tin–tungsten mineralization associated with crustal melt granites, and major orogenic gold resources; and the Shan Plateau with massive sulfide-type lead–zinc deposits [2,3].

The Southeast Asia granitoid belts, extending for 2800 km in length and up to 400 km wide, extend from Billiton Island, Indonesia, in the south, through Peninsula Malaysia via eastern Myanmar, southern peninsula Thailand to central and northern Thailand, and Yunnan Province (China) at the north. Collectively, the three granite belts of Southeast Asia (Figure 1) represent one of the greatest metallogenic provinces of the World. Their metallogenic endowment is dominated by tin–tungsten. The three granite belts are: (1) Eastern Granite Province, (2) Central Granite Province, and (3) Western Granite Province (WGP). Significant copper–gold–molybdenum porphyry-epithermal mineralization is a feature of the Central Valley Province in Myanmar [4]. The Central Granitoid Belt of Myanmar [5] approximately corresponds to WGP of the Southeast Asian granitoid tin belt [6].

In WGP, tin–tungsten mineralization is predominantly associated with Cretaceous to Eocene granites, granite pegmatites, and aplite dykes [7,8]. Over 400 tin–tungsten occurrences have been identified, including important tin–tungsten mines, such as the primary deposits at Hermyingyi, Wagone, Bawapin, Pagaye, Tagu, and the tin placer deposit at Heinda [9,10]. The tin–tungsten mineralization in the WGP occurs with cassiterite and wolframite bearing pegmatites and greisen-bordered quartz veins. These are hosted both by the granites, and also by country rocks of the Slate Belt [4]. The majority of tin–tungsten deposits are situated in the Tanintharyi (Tennanserim) Region, especially in the Dawei (Tavoy) and Myeik (Mergui) districts, except the largest deposit at Mawchi (Kayah State). The Dawei district has the largest and more important tin–tungsten lode mines, whereas more tin-rich tin–tungsten lode mines, tin lode mines, and alluvial tin deposits are found in the Myeik district [11]. The Tagu tin–tungsten deposit is located in the middle region of the WGP in the southern part of Myanmar (Figure 1), and the tin–tungsten mineralized quartz veins are hosted by both granite and metasedimentary rocks.

Although there was some literature which described the geology of the Tagu mine and the surrounding area, there were no data published with reference to the deposit geology, whole-rock geochemistry, ore mineralogy, and fluid inclusions of the Tagu deposit. In this contribution, we focus on and describe the characteristics of tin–tungsten mineralization of the Tagu deposit on the basis of field observation, geochemistry of granitic rocks, ore mineralogy, and fluid inclusion microthermometry, and discuss the origin of the fluid as well as tectonic setting of the related granitic magmatism in the area.

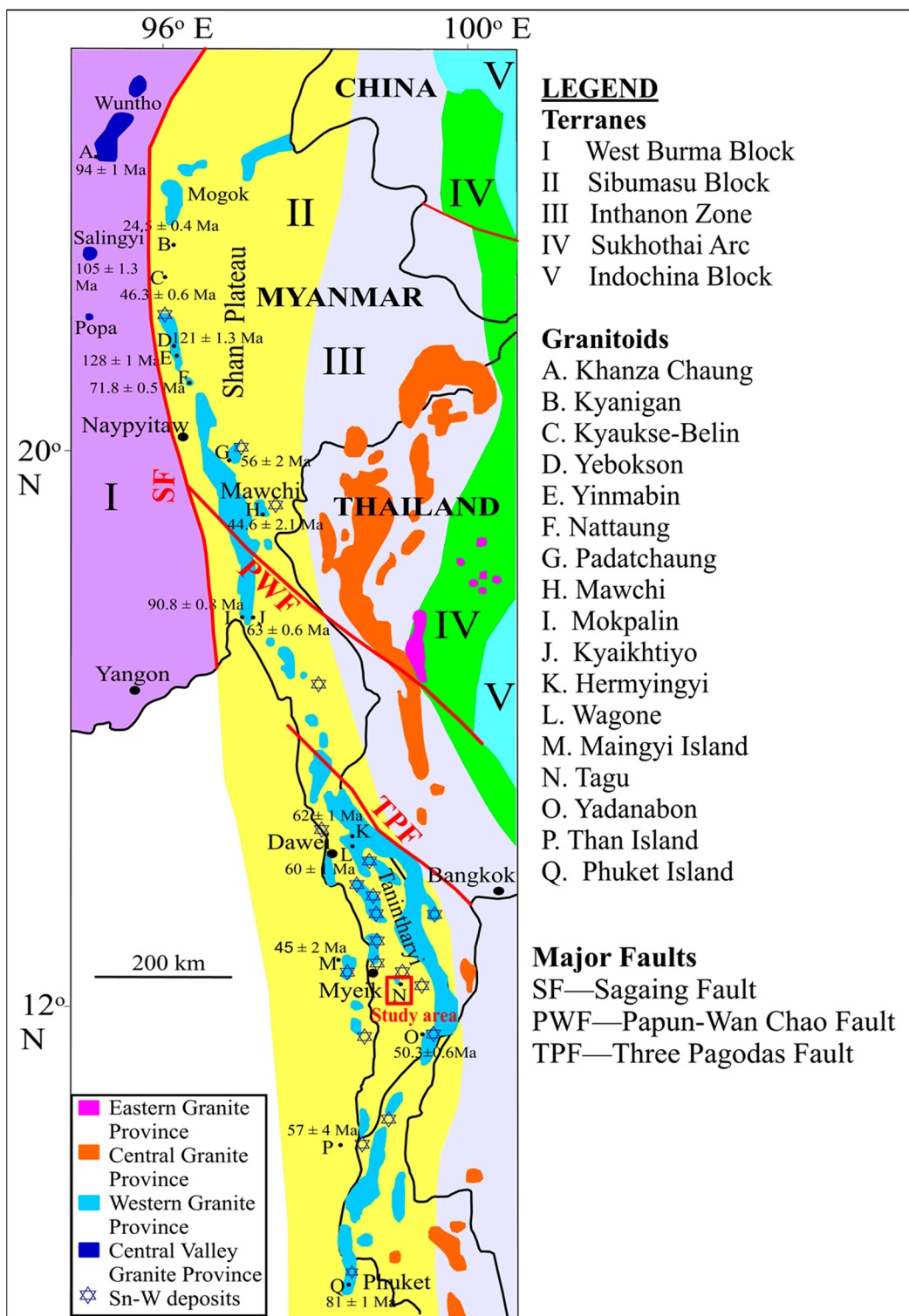

**Figure 1.** Granitoid provinces of Southeast Asia with major sutures and faults, showing locations of major Sn–W deposits and the studied area (after [6,12–14]).

## 2. Geologic Setting

### 2.1. Regional Geological Setting

Primary tin–tungsten deposits are commonly associated with the intrusion of late Cretaceous–Eocene granites of the Mogok Mandalay Mergui (MMM) Belt, which emplaced into

metasedimentary rocks of the Slate Belt [1,5,15,16]. The principal tin-producing area is located around the port town of Dawei. Much historical production was from elluvial/alluvial deposit types that are particularly focused around Myeik in the far south. This metallogenic province is also rich in tungsten spatially associated with the tin mineralization, commonly as wolframite, and more rarely as scheelite. In some deposits, tungsten contents exceed tin, and there appears to be a geographic zonation in Sn:W ratio [5,17].

Tin–tungsten mineralization is predominantly associated with granite, granitic pegmatites, and aplite dykes of the WGP, which intruded into Carboniferous to early Permian metasedimentary and sedimentary sequences of the Mergui Group [9,18–20]. The Mergui Group is mostly composed of interbedded pebbly mudstones or diamictites, with minor turbiditic wackes, thin limestones, and local white quartzites [21]. The Mergui Group is a 3000 m thick sequence of metasedimentary and sedimentary rocks comprising argillite and terrigenous clastic rocks [22]. [23] reported that granitoids in the Myeik area are distributed in three distinct ranges, namely the Frontier Range Granitoid along the Myanmar–Thai border, the Central Range Granitoid, and the Coastal Range Granitoid (Figure 2A). Most of the areas are covered by intensely deformed Mergui Group metasedimentary rocks, which trends roughly NS to NNW and is extensively intruded by granitic intrusions of various sizes (from 32 km to 64 km in length and up to 16 km in width) [24].

More than 50 tin–tungsten occurrences have been recorded in the Myeik area and the adjacent islands. The residual and detrital tin deposits frequently result from decomposition and weathering of pegmatites and tin–tungsten quartz veins. Although eluvial and alluvial cassiterite occurrences are widely scattered in the Myeik area, only some localities are important for tin–tungsten bearing pegmatites and quartz veins which have penetrated both granitoids and surrounding metasedimentary country rocks. Workable mineralized tourmaline muscovite pegmatites are noted at Kazat, Zegami, Palaw, Te-Twe, and Yengnan. Greisen-bordered tin–tungsten quartz veins are locally associated with pegmatites and, as in the Dawei area, the sub-vertical veins are found in parallel groups, mostly trending NE or sometimes E–W with steep dips. Notable tin–tungsten vein type deposits occur at Tagu (this study), Maliwun, Palauk, and Yadanabon [5].

The tin–tungsten ores in the Myeik area occur in granite, aplite dykes, pegmatites, greisens, and quartz veins. The veins frequently strike parallel to the trend of the elongated granite bodies, with a general NNW strike except for the E–W-striking vein system of the Kanbauk mine in the Dawei tin–tungsten district [9]. The texture of Mergui biotite granite varies from coarsely porphyritic to equigranular. Biotite granite consists of quartz, microcline partly sericitized, plagioclase, biotite, and minor muscovite. Tourmaline and muscovite are usual accessories found at the margins of the plutons where biotite becomes scarce. Cassiterite is reported as a minor constituent of the smaller granitoid plutons. Tin–tungsten bearing quartz veins with greisen borders and pegmatites are notably associated with the Mergui Biotite Granite. The medium-grained biotite granite appears to be a more favorable host than the coarse-grained porphyritic granite. Pegmatites occur as disconnected, lenticular veins, but trend in a regular direction parallel to the strike of the country rocks (NNW). The pegmatites contain quartz, orthoclase, and muscovite with accessory tourmaline, garnet, cassiterite, and wolframite [5].

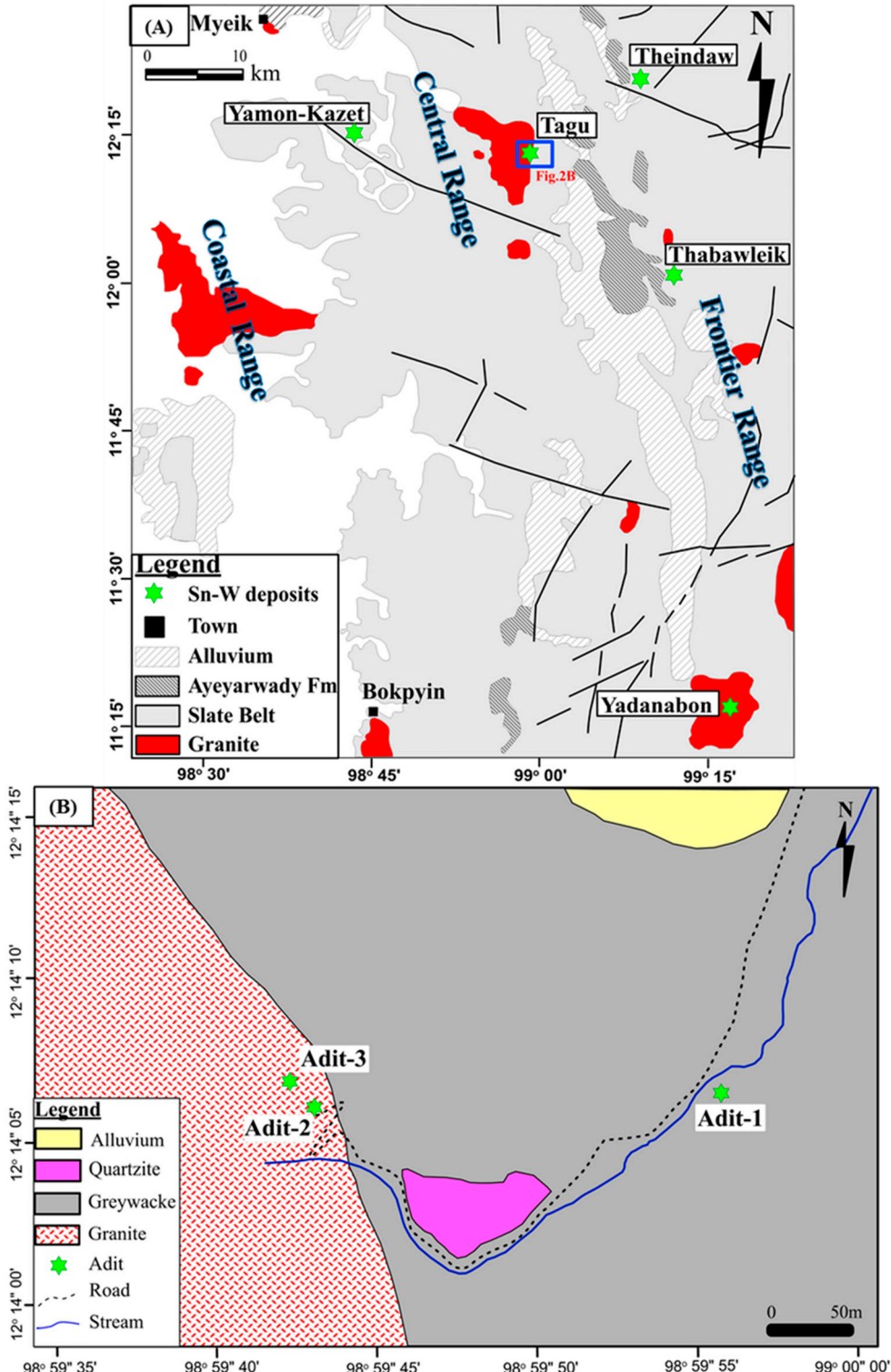

**Figure 2.** Geological setting of Myeik District, Southern Myanmar. (**A**). Regional geology of Mergui District [25]. (**B**). Geological map of the Tagu mine area.

### 2.2. Geology of Deposit

The Tagu tin–tungsten deposit, hosted in granite and Carboniferous–Permian Mergui Group metasedimentary rocks (Figure 2B) is one of the largest tin–tungsten deposits in the Myeik area. In this area, the main lithologic units are metasedimentary rocks (Mergui Group) and intrusive igneous rocks (granitic rocks), which are hosts to mineralized quartz veins. Elongated granite plutons are emplaced

along NNW–SSE, the regional strike of the Mergui Group. The width of some quartz veins in the granite are 1–5 m. More than 30 mineralized quartz veins have been observed in the metasedimentary rocks which are discordant to the direction of host rocks, while massive quartz veins were found only in the granitoid. Most mineralized quartz veins, parallel to each other, trend nearly E–W in the metasedimentary rocks. Zoned feldspar phenocrysts occur in the porphyritic granite (Figure 3A–C). The prominent rock unit of the Mergui Group is greywacke, which is well exposed along this area, as well as quartzite. Graywacke is commonly dark grey in color, fine-grained, compact, and highly jointed (Figure 3D).

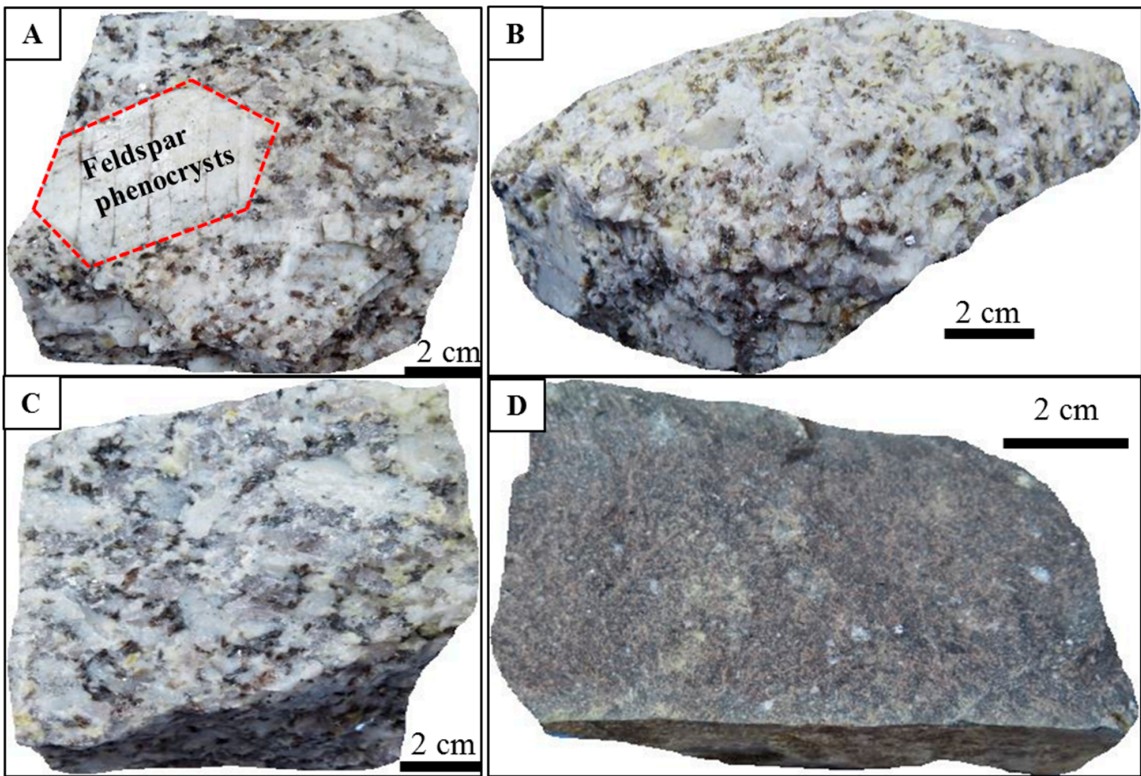

**Figure 3.** Photographs showing hand specimens of the Tagu tin–tungsten deposit. (**A**–**C**): Hand specimen of porphyritic granitic rock, which includes feldspar phenocrysts; (**D**). hand specimen of greywacke.

*2.3. Petrography of Granitoids and Greywacke*

The petrographic and textural features of the granitic rocks and greywacke are shown in Figure 4. The granitic rocks are coarse-grained, porphyritic, and have microcline, which is associated with quartz, K-feldspar, plagioclase, and biotite (Figure 4A), and secondary muscovite is present as an alteration product of K-feldspar (Figure 4B). The essential constituents are quartz, K-feldspar (orthoclase, microcline), plagioclase, biotite, and opaques (Figure 4B,C). Microcline is predominant over orthoclase. The microcline phenocrysts have inclusions of quartz and twinned plagioclase. The groundmass is composed of quartz, alkali feldspar, plagioclase, and biotite with feldspars partially sericitized (Figure 4D,E). Most of the quartz grains exhibit undulose extinction and plagioclase is highly sericitized. Greywacke of the metasedimentary sequence is fine- to medium-grained. It is mainly composed of quartz, muscovite, and iron oxide minerals. Quartz shows wavy extinction and is anhedral to polycrystalline and poorly sorted (Figure 4F).

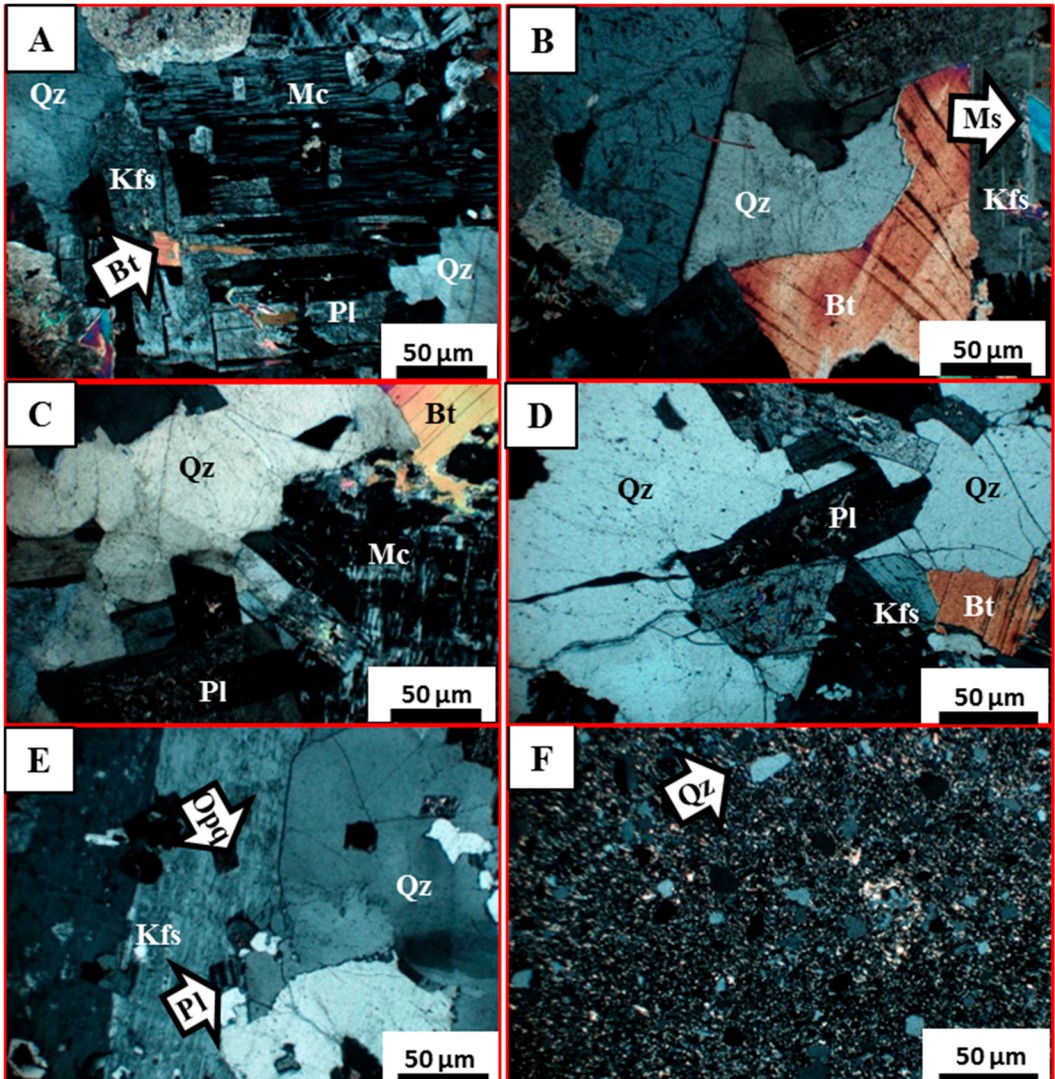

**Figure 4.** Photomicrographs of granitoids and greywacke at the Tagu tin–tungsten deposit. (**A**). Microcline associated with plagioclase and K-feldspar with a small amount of biotite in granite; (**B**). biotite associated with quartz and K-feldspar with the secondary muscovite in granite; (**C**). plagioclase associated with microcline and biotite in granite; (**D**). K-feldspar associated with biotite and plagioclase which is altered to sericite in granite; (**E**). Quartz occurring with plagioclase and K-feldspar, which has opaque minerals in granite; and (**F**). greywacke under microscope. Abbreviations: Qz = quartz, Pl = plagioclase, Kfs = K-feldspar, Bt = Biotite, Ms = muscovite, Mc = microcline, Opq = opaque mineral.

## 3. Results

### 3.1. Geochemical Characteristics of Granitic Rocks

#### 3.1.1. Analytical Techniques

A total of 30 samples of granitic rocks from the Tagu area in the Myeik District were collected. Then, 10 representative samples were selected for whole-rock geochemistry. The concentration of major and minor elements of 10 fresh granitic rocks was determined by X-ray fluorescence Spectroscopy using a RIGAKU RIX-3100 (Rigaku Corporation, Akishima City, Japan) at the Department of Earth Resources Engineering, Kyushu University. Loss on ignition (LOI) was determined by heating the samples at 1000 °C for 2 h to determine relative weight loss. The sample pellets for X-ray fluorescence (XRF)

analysis were prepared by pressing at 20 t for about 2 min in a vinyl chloride ring. The measurement conditions of analysis were 30 kv and 70 mA and the standard JA-3 andesite [26] was used to monitor the precision, yielding an error of less than 5%. The concentrations of rare earth elements (REE) were determined by inductively coupled plasma-mass spectrometry (ICP-MS) at the Center of Advanced Instrument Analysis, Kyushu University, Japan. In the preparation process, dilute HCL containing REEs was added to aluminum sec-butoxide (Al(O-sec-Bu)$_3$, TBA0 at room temperature and the solution was stirred for 2 h. Then, tetraethyl orthosilicate (Si(OEt)$_4$, TEOS) and N, N-dimethylformamid as a solvent were added into the mixed solution and stirred for 1 h at room temperature. A silica sol containing REEs surrounded by aluminums was obtained. The above reactions were carried out in a globe box replaced with N$_2$ gas and a humidity of 15% or less. To prompt hydrolysis of TEOS and polymerization of silicic acid, pure water was added to the sol solution and heated to 110 °C for 1 h. To consume the remaining HCL, propylene oxide was added. The sol solution was heated to 160 °C at a heating rate of 1 °C/h to obtain dry gel. The dry gel was heated to 1000 °C at a heating rate of 10 °C/h in an electric furnace to convert to a hard glass and stood for 2 h at 1000 °C. The concentration of REEs in the glasses was analyzed by ICP-MS after acid decomposition of the glass.

### 3.1.2. Major Oxide Elements

The SiO$_2$ contents ranged between 69.4 wt.% and 75.9 wt.%, corresponding to a felsic composition. The concentration of Al$_2$O$_3$ varied between 12.8 wt.% and 15.8 wt.%. K$_2$O ranged from 4.4 wt.% to 7.3 wt.%, Na$_2$O (from 1.6 wt.% to 2.5 wt.%), and total alkali content (Na$_2$O + K$_2$O) from 6.5 wt.% to 9.6 wt.%. MnO and P$_2$O$_5$ concentrations were less than 0.5 wt.%, and LOI values ranged from 0.8 wt.% to 1.7 wt.% (Table 1). These data were applied to construct several discrimination and variation diagrams to distinguish the rocks types, identify their geochemical characters, and constrain the tectonic setting of the granitic magmatism.

Harker diagrams (Figure 5) were used for some major oxides relative to SiO$_2$. Most of the major element oxides, NaO$_2$, MgO, TiO$_2$, Al$_2$O$_3$, K$_2$O, and CaO, were negatively correlated with SiO$_2$ (Figure 5A–F) except the slightly positively correlation with the P$_2$O$_5$ contents (Figure 5G). Classification of the rock with SiO$_2$ versus the K$_2$O + Na$_2$O geochemical rock classification diagram of [27] for plutonic rocks showed that they plotted in the granite field (Figure 6A). Almost all granite samples from the Tagu area fell in the high-potassium calc-alkaline series in the K$_2$O versus SiO$_2$ diagram (Figure 6B) [28]. The alumina saturation index (ASI) defined by molecular ratio Al$_2$O$_3$/(Na$_2$O + K$_2$O + CaO) is greater than unity (one) in all the rock samples ranging from 1.17 wt.% to 1.41 wt.%, implying that the granitic rocks were peraluminous S-type (Figure 6C) [29]. The alkali-iron-magnesium (AFM) diagram and the relative proportions of the oxides of (Na$_2$O + K$_2$O) (A), (FeO + Fe$_2$O$_3$) (F), and MgO (M) show a calc-alkaline trend (Figure 6D).

**Table 1.** Results of major (in wt.%) trace elements, including rare earth elements (in ppm) data for the Tagu granite samples, Southern Myanmar.

| Sample No. | G1 | G2 | G3 | G4 | G5 | G6 | G7 | G8 | G9 | G10 |
|---|---|---|---|---|---|---|---|---|---|---|
| Major oxide elements (wt%) | | | | | | | | | | |
| $SiO_2$ | 71.53 | 73.56 | 71.65 | 71.79 | 72.34 | 71.15 | 71.15 | 69.48 | 71.68 | 75.98 |
| $TiO_2$ | 0.31 | 0.38 | 0.32 | 0.36 | 0.30 | 0.27 | 0.31 | 0.31 | 0.36 | 0.30 |
| $Al_2O_3$ | 13.67 | 13.68 | 14.95 | 14.72 | 14.45 | 15.06 | 15.09 | 15.88 | 14.36 | 12.83 |
| FeO | 2.08 | 2.27 | 2.07 | 2.08 | 2.06 | 1.82 | 1.88 | 1.58 | 2.18 | 1.68 |
| MnO | 0.04 | 0.04 | 0.04 | 0.04 | 0.03 | 0.03 | 0.06 | 0.02 | 0.04 | 0.06 |
| MgO | 0.71 | 0.79 | 0.77 | 0.78 | 0.67 | 0.61 | 0.65 | 0.64 | 0.72 | 0.57 |
| CaO | 1.23 | 1.25 | 1.11 | 1.00 | 1.05 | 1.04 | 1.16 | 0.71 | 1.18 | 0.61 |
| $Na_2O$ | 1.62 | 2.19 | 2.28 | 2.07 | 2.50 | 2.39 | 2.37 | 2.30 | 2.24 | 1.89 |
| $K_2O$ | 6.70 | 4.61 | 5.01 | 5.02 | 4.46 | 5.92 | 5.49 | 7.37 | 5.59 | 4.62 |
| $P_2O_5$ | 0.16 | 0.20 | 0.18 | 0.17 | 0.17 | 0.14 | 0.15 | 0.18 | 0.17 | 0.15 |
| LOI | 1.74 | 0.86 | 1.42 | 1.78 | 1.51 | 1.35 | 1.51 | 1.30 | 1.27 | 1.09 |
| Total | 99.79 | 99.83 | 99.80 | 99.81 | 99.54 | 99.79 | 99.81 | 99.78 | 99.78 | 99.77 |
| Trace elements (ppm) | | | | | | | | | | |
| Sc | 1.3 | 0.8 | 1.0 | 2.2 | 1.7 | 2.1 | 2.1 | 1.8 | 1.7 | 1.9 |
| V | 31 | 36 | 24 | 25 | 28 | 17 | 25 | 27 | 31 | 21 |
| Co | 45 | 30 | 40 | 46 | 38 | 30 | 38 | 41 | 29 | 26 |
| Ni | 15 | 15 | 11 | 12 | 15 | 13 | 16 | 15 | 14 | 13 |
| Cu | 23 | n.d | 6 | 64 | 34 | 14 | 11 | 3 | 4 | 97 |
| Zn | 91 | 12 | 203 | 154 | 81 | 74 | n.d | 23 | 153 | 61 |
| Pb | 44 | 24 | 34 | 65 | 31 | 57 | 52 | 98 | 70 | 109 |
| As | 1 | 9 | 7 | n.d | 7 | n.d | n.d | n.d | 95 | 148 |
| Mo | 12 | 16 | 18 | 15 | 13 | 11 | 11 | 14 | 18 | 15 |
| Rb | 911 | 633 | 634 | 621 | 680 | 801 | 846 | 757 | 727 | 835 |
| Sr | 32 | 31 | 35 | 36 | 26 | 40 | 46 | 66 | 49 | 25 |
| Ba | 349 | 202 | 250 | 264 | 159 | 257 | 318 | 437 | 320 | 234 |
| Y | 31 | 30 | 29 | 29 | 28 | 33 | 33 | 31 | 31 | 35 |
| Zr | 188 | 204 | 197 | 200 | 154 | 158 | 181 | 192 | 210 | 217 |
| Nb | 28 | 32 | 27 | 29 | 32 | 27 | 36 | 23 | 29 | 27 |
| Th | 56 | 61 | 58 | 58 | 47 | 44 | 46 | 49 | 58 | 53 |
| U | 15 | 38 | 44 | 21 | 31 | 19 | 22 | 21 | 13 | 15 |
| Sb | 18 | 20 | 14 | 24 | 19 | 18 | 15 | 16 | 21 | 15 |
| Sn | 114 | 57 | 50 | 56 | 63 | 53 | 72 | 49 | 59 | 152 |
| W | 36 | 32 | 51 | 60 | 48 | 35 | 349 | 8 | 23 | 35 |
| Rare earth elements (ppm) | | | | | | | | | | |
| La | 17.4 | 25.7 | 15.7 | 23.2 | 20.7 | 19.0 | 17.4 | 12.4 | 26.2 | 23.3 |
| Ce | 36.7 | 80.7 | 34.6 | 76.1 | 46.7 | 39.4 | 38.4 | 27.6 | 82.3 | 73.6 |
| Pr | 4.1 | 6.3 | 3.9 | 5.8 | 5.1 | 4.6 | 4.1 | 3.0 | 6.4 | 5.6 |
| Nd | 14.7 | 22.3 | 14.2 | 20.9 | 18.2 | 16.5 | 15.2 | 11.1 | 23.3 | 20.1 |
| Sm | 2.5 | 4.0 | 2.5 | 3.7 | 3.2 | 3.0 | 2.6 | 2.1 | 4.2 | 3.6 |
| Eu | 0.2 | 0.2 | 0.2 | 0.2 | 0.2 | 0.2 | 0.2 | 0.1 | 0.1 | 0.2 |
| Gd | 1.4 | 2.4 | 1.4 | 2.1 | 1.9 | 1.7 | 1.4 | 1.2 | 2.5 | 2.0 |
| Tb | 0.0 | 0.1 | 0.0 | 0.1 | 0.1 | n.d | n.d | n.d | 0.1 | 0.1 |
| Dy | 0.3 | 0.6 | 0.4 | 0.6 | 0.5 | 0.5 | 0.3 | 0.4 | 0.7 | 0.5 |
| Ho | 0.1 | 0.1 | 0.1 | 0.1 | 0.1 | 0.1 | 0.1 | 0.1 | 0.1 | 0.1 |
| Er | 0.0 | 0.2 | n.d | 0.1 | 0.1 | 0.1 | n.d | n.d | 0.1 | 0.1 |
| Tm | 0.1 | 0.1 | 0.1 | 0.1 | 0.1 | 0.1 | 0.1 | 0.1 | 0.1 | 0.1 |
| Yb | 0.2 | 0.3 | 0.2 | 0.2 | 0.3 | 0.2 | 0.2 | 0.2 | 0.3 | 0.2 |
| Lu | 0.1 | 0.1 | 0.1 | 0.1 | 0.1 | 0.1 | 0.1 | 0.1 | 0.1 | 0.1 |
| Total REE | 77.8 | 143.1 | 73.4 | 133.3 | 97.3 | 85.5 | 80.1 | 58.4 | 146.5 | 129.6 |

* n.d = not detected.

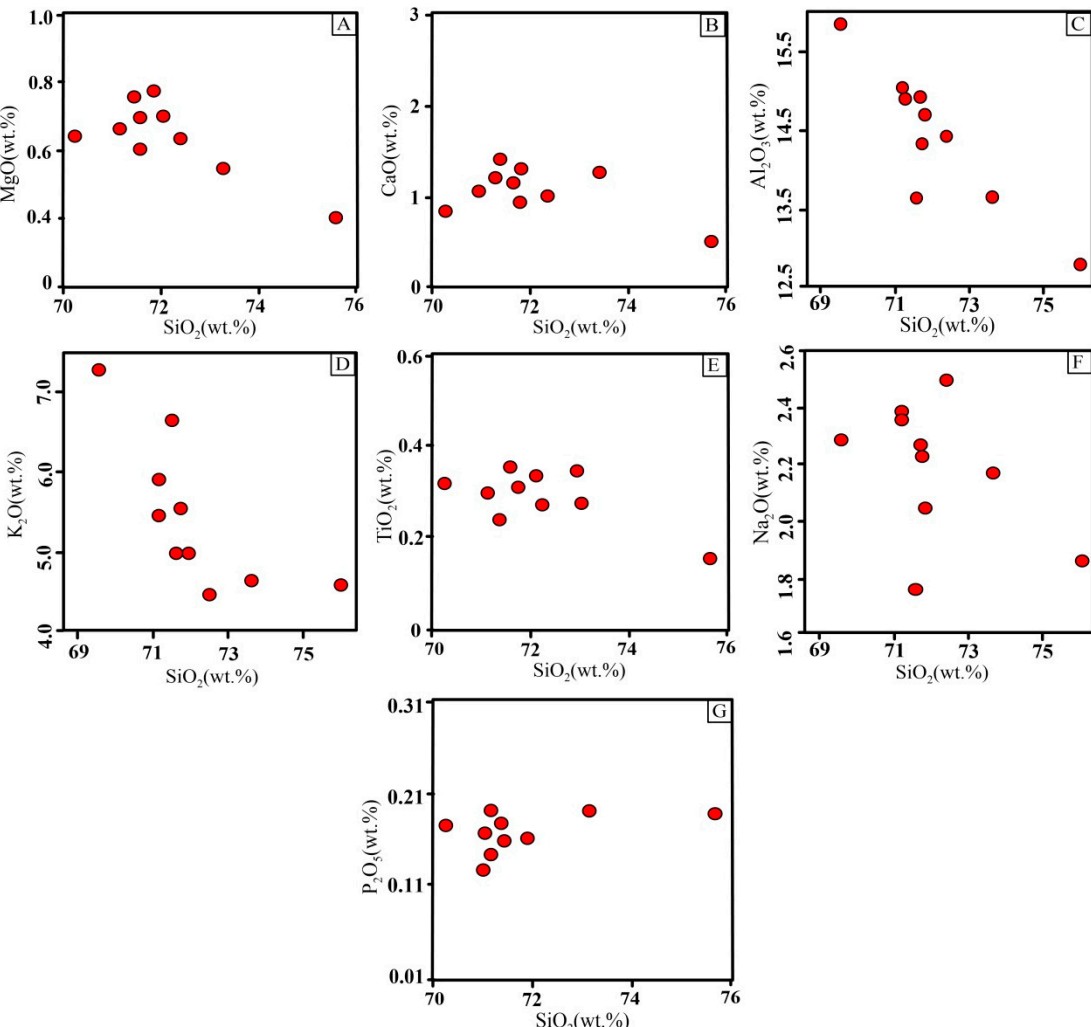

**Figure 5.** Harker variation diagrams; silica (SiO$_2$ wt.%) plotted against a range of major oxides (in wt.%) for the granites of the Tagu area. (**A**). MgO is negatively correlated with SiO$_2$; (**B**). CaO is negatively correlated with SiO$_2$; (**C**). Al$_2$O$_3$ is negatively correlated with SiO$_2$; (**D**). K$_2$O is negatively correlated with SiO$_2$; (**E**). TiO$_2$ is negatively correlated with SiO$_2$; (**F**). NaO$_2$ is negatively correlated with SiO$_2$; (**G**). P$_2$O$_5$ is slightly positively correlated with SiO$_2$.

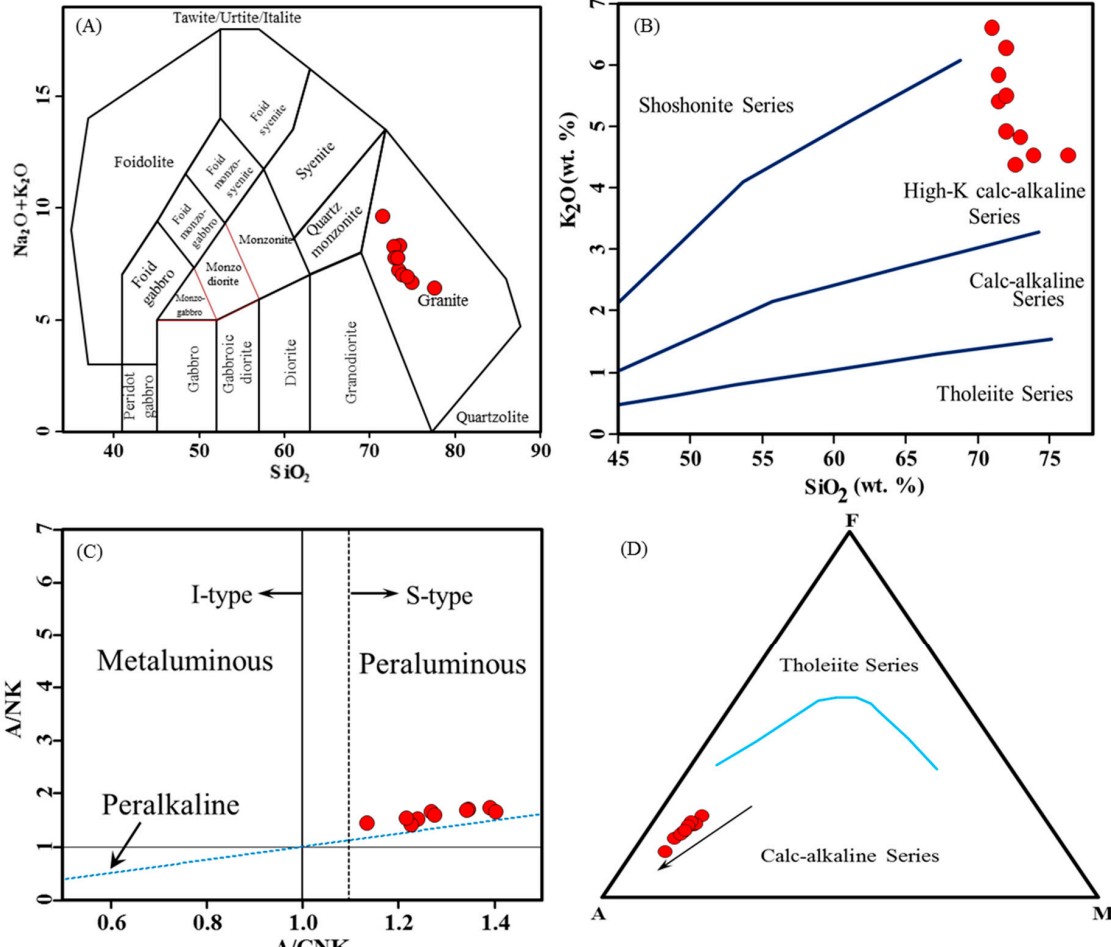

**Figure 6.** Oxide element geochemical data of granitic rocks from the Tagu area. (**A**). SiO$_2$ versus Na$_2$O + K$_2$O binary diagram shows that the majority of samples fell in the granitoid field [27]; (**B**). K$_2$O wt.% versus SiO$_2$ wt.%; the subdivisions are by [28]; (**C**). Alumina saturation index diagram: The A/CNK (molar Al$_2$O$_3$/(CaO + Na$_2$O + K$_2$O)) ratio is greater than 1.1 and it can be assumed that the granite is peraluminous and S-type [29]; (**D**). AFM triangular diagram for granitoids after [30]. A = (Na$_2$O + K$_2$O) wt.%, F = (FeO + Fe$_2$O$_3$) wt.%, and M = MgO wt.%.

### 3.1.3. Trace Elements

Trace element concentrations of the granite rocks from the Tagu are listed in Table 1. The concentrations of large ion lithophile elements, such as Ba (159–437 ppm), Rb (621–911 ppm), Sr (25–49 ppm), and Pb (31–109 ppm), were high. It was noted that the high field strength elements (HFSE), such as Zr (154–217 ppm), Nb (23–36 ppm), Y (28–35 ppm), Th (44–61 ppm), and U (13–44 ppm) were also enriched. The ternary diagram Rb–Ba–Sr was applied to understand the processes for both the classification and genetic types of plutonic rocks [31] (Figure 7A). Trace element discrimination (Y + Nb) versus Rb diagrams [32] indicated that the granites of the Tagu area fall in the syn-collisional setting (Figure 7B).

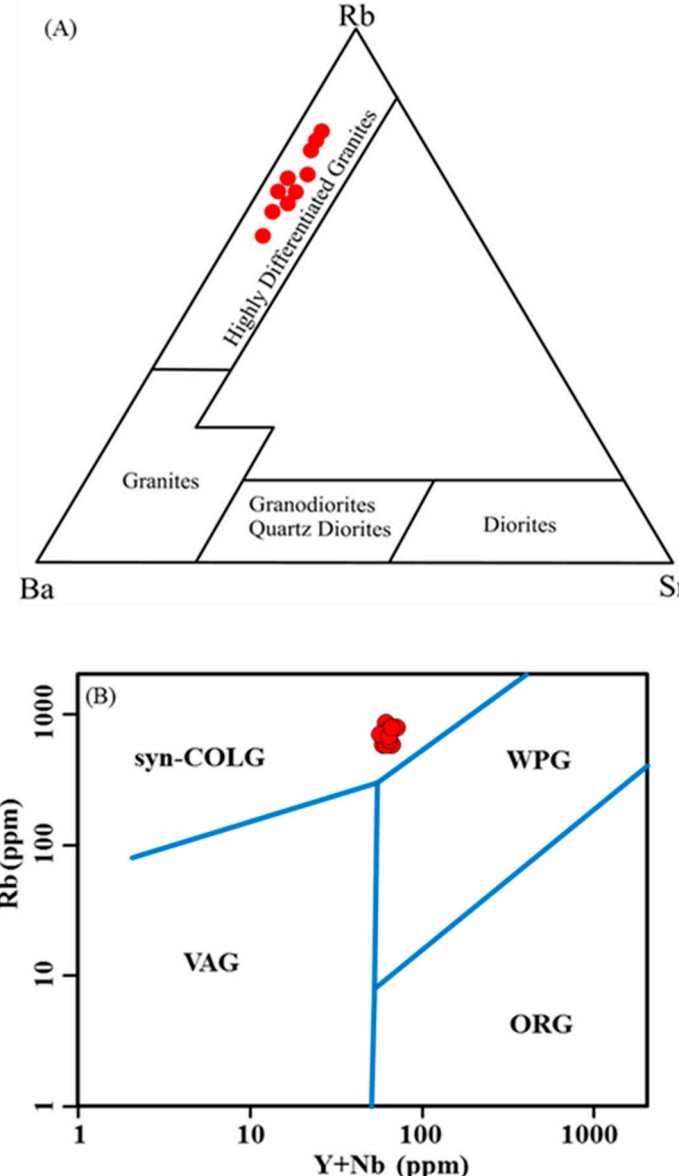

**Figure 7.** Rb–Sr–Ba triangular diagram. (**A**). Rb–Sr–Ba triangular plot pointing out the genetic aspect of the Tagu granitoid [31]; (**B**). Y + Nb versus Rb diagram showing the tectonic setting [32] of the Tagu granitic rocks. Syn-COLG = syncollisional granites; VAG = volcanic-arc granites; WPG = within-plate granites; ORG = oceanic ridge granites.

The concentrations of light rare earth elements (LREE) of granitic rocks are generally elevated (La: 12.4–26.2 ppm, Ce: 27.6–82 ppm, Nd: 11.1–23.3 ppm, and Sm: 2.1–4.2 ppm) in contrast to the depleted heavy rare earth elements (HREE). The chondrite normalized REE pattern of granites (Figure 8A) showed a significant negative Eu anomaly. A primitive mantle-normalized spider diagram of [33] for the granitic rock samples (Figure 8B) reveals the enrichment of large-ion lithophile elements (LILEs), such as Rb, Pb, and K, exhibiting distinct negative anomalies for high-field-strength elements (HFSEs), such as Nb, P, and Ti, and negative anomalies of Ba and Sr from Rb, which are remarkable.

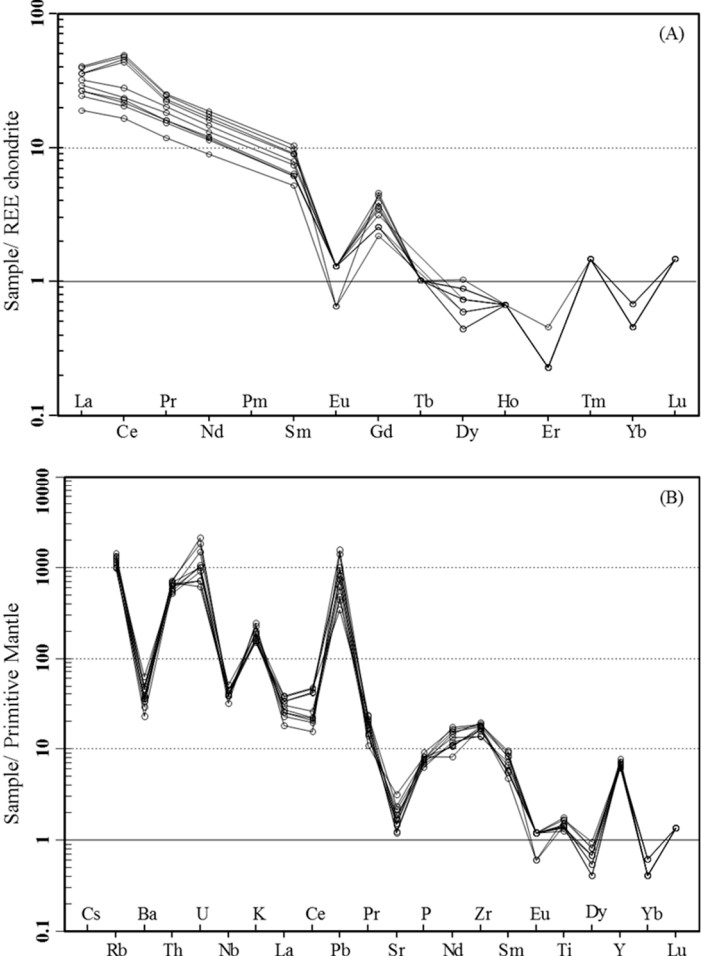

**Figure 8.** Chondrite normalized rare earth elements (REE) diagram (**A**) and primitive mantle normalized trace element diagram (**B**) for Tagu granitoids (normalizing values from [33]).

### 3.2. Mineralization of Tagu Sn–W Deposit

The major mineralization style of the Tagu deposit is represented by cassiterite–wolframite bearing quartz veins that are hosted in the granitic rocks and metasedimentary rocks. The steeply dipping mineralized quartz veins contain oxide ore minerals, such as cassiterite and wolframite, which are associated with sulfide minerals. Some mineralized quartz veins parallel to each other and trending nearly E–W directions have a pinch and swell structure and broken quartz veins by the faults during tectonic events (Figure 9A,B). The mineralized quartz veins are pinched out and branched upwards. Subsequently, these veins are terminated as small parallel and branched veins into the metasedimentary rocks (Figure 9C,D). The ore minerals are mostly associated with quartz and muscovite. Almost all of the quartz veinlets are parallel to each other. There are up to 30 mineralized quartz veins trending nearly E–W and steeply dipping in the metasedimentary rocks, whereas in the granite, the mineralized massive quartz veins are 1 m to 5 m across and trend nearly NEE.

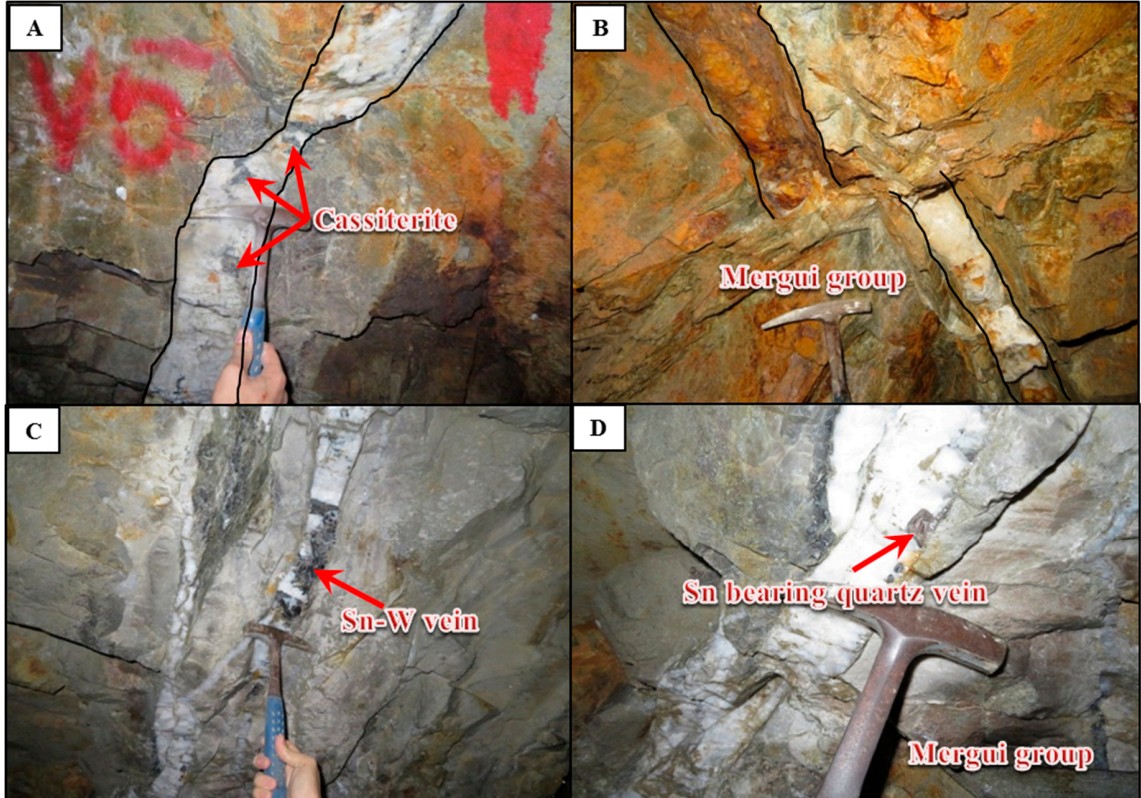

**Figure 9.** Field photos for lithologic units in the Tagu area. (**A**). Cassiterite bearing quartz vein which has a pinch and swell structure; (**B**). broken quartz vein in the metasedimentary rocks by faults; (**C**). Sn–W-bearing quartz vein in metasedimentary rocks; (**D**). Sn-bearing quartz vein within the metasedimentary rock.

*3.3. Mineralogy and Paragenesis*

The three ore formation stages are characterized by oxide ore stage, sulfide stage, and supergene stage in the Tagu tin–tungsten deposit. Early formed minerals are characterized by major oxide ore minerals, such as cassiterite and wolframite, followed by sulfide minerals. In the oxide ore stage, the deposition of wolframite appears somewhat later than cassiterite. Cassiterite occurs as idiomorphic crystals as well as mixed with wolframite (Figure 10A). Most cassiterites are deformed and replaced by wolframite and sulfide minerals, such as sphalerite and chalcopyrite (Figure 10B,C).

Some sulfide minerals fill the fracture within cassiterite and wolframite. This suggests that cassiterites and wolframite were formed earlier than other sulfide minerals. Most sulfide minerals from the sulfide-bearing vein coexist with mutual intergrowth (Figure 10D–F). Covellite from the sulfide-bearing vein replaced the rims of chalcopyrite along the grain boundaries (Figure 10G–I). Pyrite is associated with chalcopyrite and arsenopyrite and oxide minerals. It has mutual intergrowth with other sulfide minerals, such as chalcopyrite, sphalerite, and molybdenite. Pyrite is most commonly found as a massive aggregate, but sometimes as veinlets within the massive wolframite. Sphalerite is found as anhedral, disseminated grains, or associated with pyrite, chalcopyrite, and molybdenite.

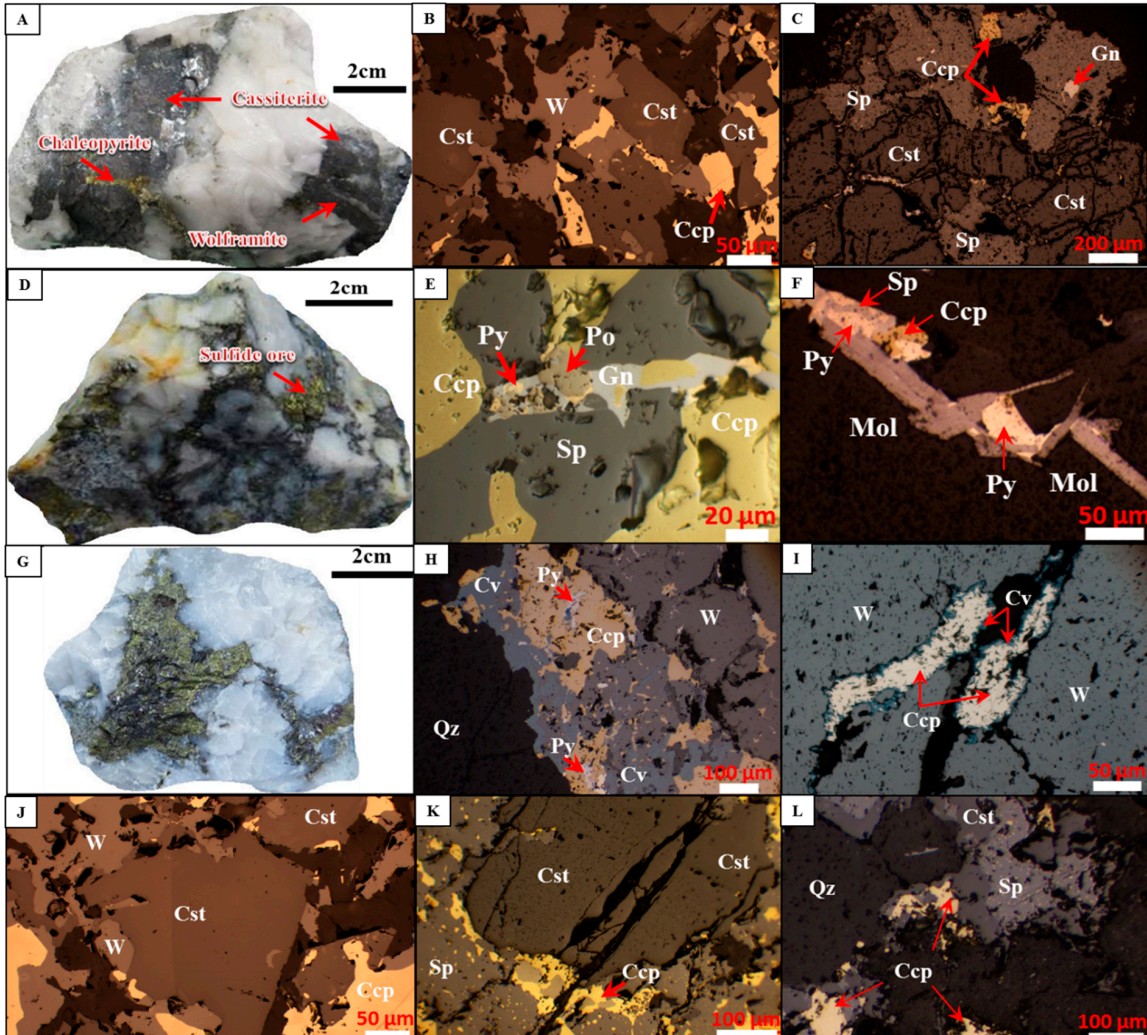

**Figure 10.** Photographs of Sn–W ore specimen and ore microscopic studies. (**A**). Cassiterite and wolframite mineralized quartz vein. (**B**). Cassiterite replaced by wolframite, which has fracture filling of chalcopyrite. (**C**). Sphalerite with chalcopyrite blebs replacing between the fracture of cassiterite. (**D**). Sulfide-bearing quartz vein. (**E**). Sulfide ores are coexisting and in mutual intergrowth with each other. (**F**). Molybdenite associated with other sulfide minerals. (**G**). Sulfide-bearing quartz vein. (**H**). Covellite is observed as rims replacing chalcopyrite. (**I**). Chalcopyrite with covellite rims as fracture filling between the fracture of wolframite. (**J**). Cassiterite replaced by wolframite associated with chalcopyrite. (**K**). Cassiterite replaced by sphalerite with chalcopyrite blebs. (**L**). Cassiterite replaced by sphalerite and chalcopyrite. Abbreviations: Qz = quartz, Cst = cassiterite, W = wolframite, Py = pyrite, Ccp = chalcopyrite, Po = pyrrhotite, Sp = sphalerite, Gn = galena, Cv = covellite, Mol = molybdenite.

Cassiterite and wolframite are closely associated with sphalerite in the quartz veins, both in granite and metasedimentary rocks. Sphalerite occurred as replacement and fracture filling mineral along the grain boundaries and fractures of cassiterite and wolframite crystals (Figure 10J–L). Sometimes it enclosed chalcopyrite blebs. Consequently, it is assumed that sphalerite occurs contemporaneously with chalcopyrite. According to ore microscopic studies, the paragenetic sequence representing only the tin–tungsten vein mineralization at the Tagu deposit was constructed (Figure 11).

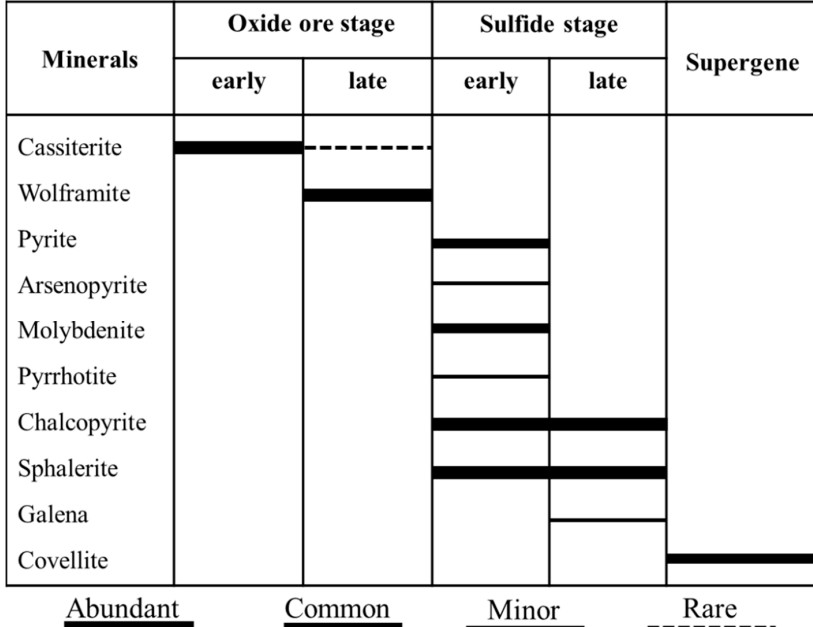

**Figure 11.** Paragenetic sequence of ore mineralization of the Tagu Sn–W deposit, southern Myanmar.

*3.4. Fluid Inclusion Studies*

The homogenization temperatures of about 300 fluid inclusions in seven vein quartz samples were determined in this present study. The detailed microthermometric data are presented mainly for fluid inclusions in veined quartz, hosted in granite, and metasedimentary rocks.

3.4.1. Method

Fluid inclusion microthermometry on doubly polished sections was conducted at the Department of Earth Resources Engineering, Kyushu University, Japan. Microthermometric analyses were carried out using a Linkam LK600 heating–freezing stage with a Nikon Y-IM microscope. The errors for the temperature measurements were ±0.5, ±0.2, and ±2.0 °C for runs in the range of −120 °C to −70 °C, −70 °C to 100 °C, and 100 °C to 600 °C, respectively. The final ice-melting temperatures were measured at a heating rate of less than 0.1 °C/min and the homogenization temperatures at a rate of ≤1 °C/min. Total salinities for NaCl–H$_2$O fluid inclusions were calculated from the final ice melting temperatures using the equation of wt.% NaCl = 1.78 T − 0.0442 T$^2$ + 0.000557 T$^3$, where T is depression temperature in °C [34], and is expressed as wt.% NaCl equivalent. The salinities of the CO$_2$–H$_2$O inclusions were calculated from the melting temperature of clathrate using the equation wt.% NaCl = 15.52022 − 1.02342T − 0.05286T$^2$, where T is the clathrate melting temperature expressed in °C [35,36].

3.4.2. Fluid Inclusion Petrography

The pattern, shape, size, and phase content of fluid inclusions within the quartz crystals were examined to determine the fluid inclusion types and to distinguish primary and secondary inclusions, as described by [37]. The shapes of fluid inclusions are variable, exhibiting spheroidal, irregular, polygonal, rounded, sub-rounded, and elongated. The size of fluid inclusions ranges from 5 µm to 30 µm in diameter. The fluid inclusions are commonly distributed in clusters, in isolation, and along growth zones, and are considered to be primary. According to the phase relations observed at room temperature, fluid inclusions from the Tagu Sn–W deposit were classified into three major types.

Type-A: This type of fluid inclusion was liquid-dominated two-phase, consisting of liquid and vapor (L + V) at room temperature, occurring as isolated inclusions, clusters, and planes of inclusions in the milky quartz of tin–tungsten and sulfide mineralized veins (Figure 12A–C). The V/(L + V) ratio

was 0.15–0.35 for type A. They ranged in size from 5 μm to 20 μm and were rounded, sub-rounded, elongated, and irregularly-shaped. They were a commonly found fluid inclusion type in the tin–tungsten mineralized vein quartz in metasedimentary rocks and also granite.

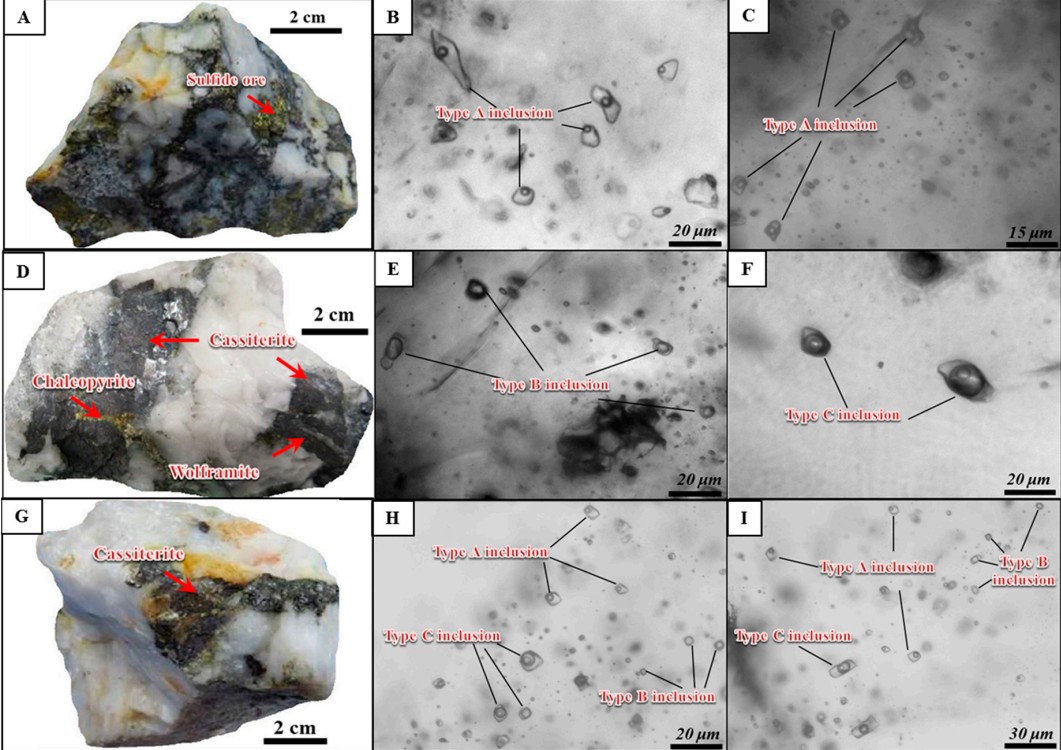

**Figure 12.** Photomicrographs of fluid inclusion types of the Tagu Sn–W deposit. (**A**). Sulfide-bearing quartz vein. (**B**). Type-A fluid inclusions in metasedimentary hosted quartz. (**C**). Type-A fluid inclusions in granite-hosted quartz. (**D**). Sn–W bearing quartz vein from granite rock. (**E**). Type-B fluid inclusions in the granite-hosted quartz. (**F**). Type-C fluid inclusions in granite-hosted quartz vein. (**G**). Sn–W bearing quartz vein from granite rock. (**H**). Type-A, B, and C fluid inclusions coexisting together within a small area in granite-hosted quartz. (**I**). Type-A, B, and C fluid inclusions coexisting together within a small area of the sample.

Type-B: This fluid inclusion type was vapor-dominated two-phase, consisting of vapor and liquid (V + L) at room temperature occurring as isolated inclusions, clusters, and planes of inclusions in the milky quartz of tin–tungsten mineralized veins in granite (Figure 12D–F). The V/(L + V) ratio was 0.3–0.8 for type B. They ranged in size from 5 μm to 25 μm and were spheroidal, irregular polygonal, and irregularly-shaped. They were the most common inclusion type in the quartz of tin–tungsten mineralized veins in granite.

Type-C: At room temperature these fluid inclusions consisted of three phases, aqueous liquid + liquid $CO_2$ + vapor $CO_2$, with the volume of the $CO_2$ phase (liquid) varying widely from 60 vol % to 85 vol % which is found in Sn-W mineralized veins in granite (Figure 12G–I). They occurred in planes of growth zones in vein quartz. They ranged in size from 10 μm to 30 μm and were in spheroidal, irregular polygonal, and irregular shapes. They were observed only in the quartz of the Sn–W mineralized veins in granite.

3.4.3. Results of Microthermometric Measurements

The microthermometric data of type-A, type-B, and type-C inclusions are listed in Table 2. For (type-A) inclusions, final ice melting temperatures (Tice) varied from −0.6 °C to −5.8 °C for the vein quartz in metsedimentary rocks, and −1.7 °C to −7.1 °C for vein quartz in the granite,

with corresponding salinities from 1.1 wt.% to 8.9 wt.% NaCl equivalent and 2.9 wt.% to 10.6 wt.% NaCl equivalent, respectively. The homogenization temperatures of type-A fluid inclusions varied from 140 °C to 330 °C (peak at 230 °C) for the vein quartz in metasedimentary rocks, and from 230 °C to 370 °C (peak at 280 °C) for quartz veins in granite (Figure 13A,B), lower than the total homogenization temperature (Th-tot) of type-C fluid inclusions.

**Table 2.** Summary of fluid inclusion types and microthermometric data of the Tagu Sn–W deposit, Southern Myanmar.

| Mineral | Type | Tm,CO$_2$ (°C) | Tm,clath (°C) | Th,CO$_2$ (°C) | Th (°C) | Tice (°C) | FI (N) | Salinity (wt.% NaCl$_{eqv.}$) |
|---|---|---|---|---|---|---|---|---|
| Granite hosted quartz | Type-A | | | | 230~370 | −1.7~−7.1 | N = 89 | 2.9~10.6 |
| | Type-B | | | | 310~390 | −4.2~−8.4 | N = 34 | 6.7~12.2 |
| | Type-C | −59.5~−56.8 | 7.1~9.1 | 28.3~31.1 | 270~405 | | N = 61 | 1.8~5.6 |
| Meta-sedimentary hosted quartz | Type-A | | | | 140~330 | −0.6~−5.8 | N = 113 | 1.1~8.9 |

Tm,CO$_2$—final melting temperature of CO$_2$; Tm,clath—final melting temperature of CO$_2$ clathrate; Th,CO$_2$—partial homogenization temperature of CO$_2$ inclusions; Th—total homogenization temperature of inclusions; Tice—final ice melting temperature, FI(N)—fluid inclusion number.

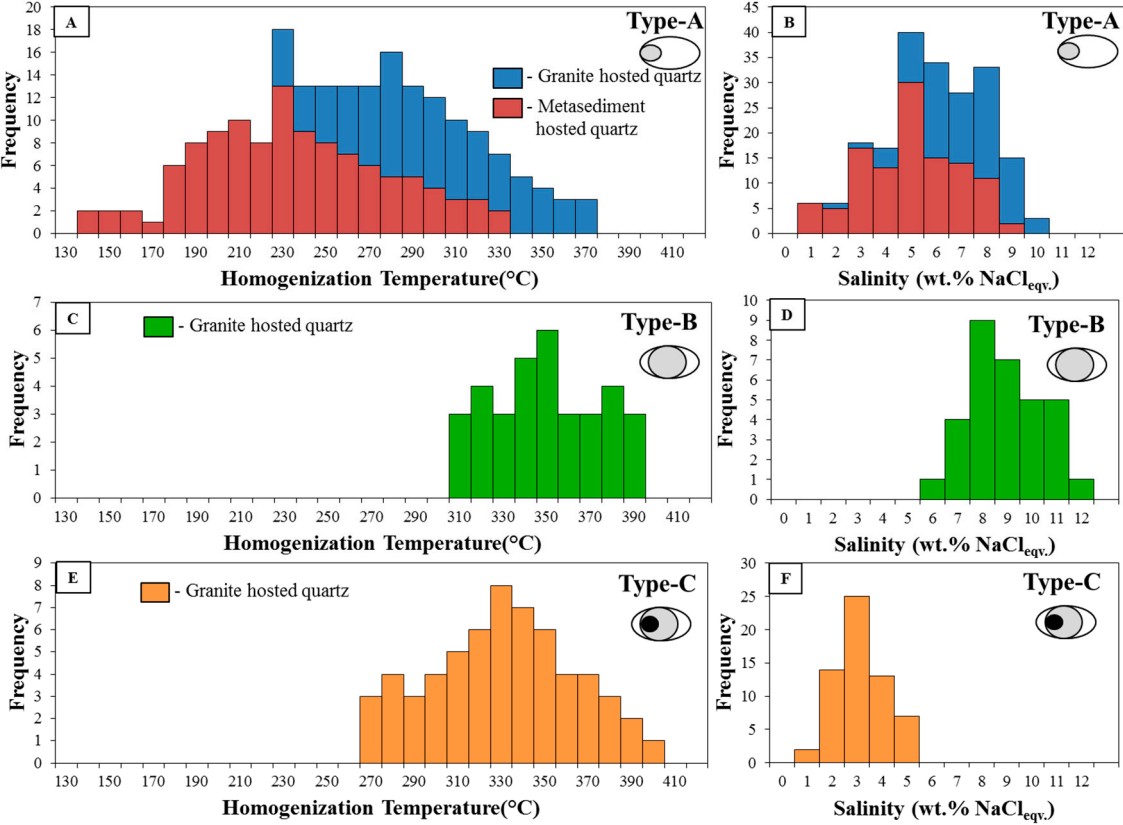

**Figure 13.** Histograms of microthermometric data for fluid inclusions in the Tagu Sn–W deposit. (**A**). Homogenization temperature of type-A fluid inclusions from granite and metasedimentary hosted quartz veins; (**B**). Salinity of type-A fluid inclusions from granite and metasedimentary hosted quartz veins; (**C**). Homogenization temperature of type-B fluid inclusions from granite hosted quartz veins; (**D**). Salinity of type-B fluid inclusions from granite hosted quartz veins; (**E**). Homogenization temperature of type-C fluid inclusions from granite hosted quartz veins; (**F**). Salinity of type-C fluid inclusions from granite hosted quartz veins.

The type-B vapor-rich inclusions homogenized to the liquid phase from 310 °C to 390 °C (peak at 350 °C) and their Tm-ice ranged from −4.2 °C to −8.4 °C, which corresponds to salinities from 6.7 wt.% to 12.2 wt.% NaCl equivalent (Figure 13C,D).

The $CO_2$ melting temperatures (Tm-$CO_2$) of $CO_2$-bearing inclusions in mineralized quartz veins in the granite ranged from −59.5 °C to −56.8 °C, suggesting that the carbonic phase was dominated by $CO_2$, with only minor components of other volatiles, such as $CH_4$ and $N_2$ [38]. The partial homogenization temperatures (Th-$CO_2$) of the $CO_2$ phase into the liquid phase ranged from 28.3 °C to 31.1 °C. The total homogenization temperatures (Th-tot) ranged from 270 °C to 405 °C (mode at 330 °C). The clathrate melting temperatures (Tm-clath) of type-C fluid inclusions varied from 7.1 °C to 9.1 °C, corresponding to salinities ranging from 1.8 wt.% to 5.6 wt.% NaCl equivalent (Figure 13E,F). The type-C, $CO_2$-bearing inclusions were marked by relatively high total homogenization temperatures and lower salinities than the type-A liquid-rich inclusions and type-B, vapor-rich inclusions (Figure 14).

## 4. Discussions

### 4.1. Petrogenesis of Granitic Rocks

Strongly peraluminous S-type granite formed from the partial melting of peraluminous clastic sediments is common in continental collisional zones, even in Neoarchean time [39]. The granite of the Tagu tin–tungsten deposit is peraluminous, which is characterized by the A/CNK (molar $Al_2O_3$/(CaO + $Na_2O$ + $K_2O$)) value ranging from 1.17 to 1.41. The $SiO_2$ content is negatively correlated with $Na_2O$, MgO, $TiO_2$, $Al_2O_3$, $K_2O$, and CaO contents (Figure 5). The content of $P_2O_5$ in highly fractionated S-type granites was high compared to I-type granites [40,41]. In the Tagu tin–tungsten deposit, the $P_2O_5$ contents of biotite granite were slightly positively correlated with the content of $SiO_2$. A highly fractionated S-type granitic magma was revealed (Figure 5) and the relationship between Rb–Ba–Sr also exhibited highly fractionated granite (Figure 7A).

The granite at Tagu is characterized by generally elevated REE contents with negative Eu anomaly, which can be produced by either partial melting of a plagioclase-rich source or fraction of plagioclase, or a combination of both [42]. According to the primitive mantle-normalized spider diagram of [33], the granite at Tagu (Figure 8B) revealed an enrichment of LILEs, such as Rb, K, and Pb, exhibiting distinct negative anomalies of HFSEs, such as Nb, P, and Ti. Most of these features, such as negative Ba, Sr, Nb, and Ti anomalies and positive Rb, Th, and La anomalies, are compatible to those of typical crustal melt, e.g., S-type granites in North Queensland [41] and the Himalayan leucogranite [43]. Furthermore, enrichment of Rb, Th, and Y is also consistent with crustal derivation. Thus, parental magma of the granites at Tagu may have been derived from the crustal source.

### 4.2. Tectonic Setting

The tin granites of WGP have been shown to have been derived from the partial melting of the crust [3,44] and the emplacement of these granites has been variously related to the Cretaceous–Paleogene subduction and Himalayan orogeny-related collision [45], back-arc extension, or post-collisional, syncollisional extension related granitic magmatism [21,46–48]. [3] considered that tin–tungsten mineralized granites of the Mogok–Mandalay–Mergui (MMM) Belt intruded into the Slate Belt (e.g., at Hermyingyi, Mawchi, and Yadanabon). The zircon U–Pb ages of both mineralized and presumed coeval non-mineralized granites within the MMM Belt are largely of Palaeogene age [3]. The zircon U–Pb ages of 75–50 Ma from granites within the southern MMM Belt suggest that the collision occurred during Late Cretaceous [3]. [49] also noted similar collisional setting from the metallogeny of the Central Andean Tin Belt. [8] proposed that the Slate Belt represents a separate continental fragment, which was derived from the Gondwana margin and was accreted onto the western Sibumasu margin, probably during the Jurassic period [50,51]. There is an agreement that by the Early Cretaceous period the subduction of the Mesotethys Ocean beneath the western margin of the Slate Belt–Sibumasu Block was responsible for the growth of a continental-margin plutonic

arc. This plutonic arc was described by [6] as the "Western Granite Province of Southeast Asia" and involved the emplacement of both I- and S-type granites during the Early and Late Cretaceous–Eocene period. S-type granites occur in various settings that are generally related to plate convergence. Perhaps the best constrained are the S-type granites that clearly formed because of the collision of India and Asia [52]. Moreover, many S-types are synkinematic. These facts led to the notion that S-type granites are syn-collisiorial granites [52], as in the geochemical classification of [32]. The granite of the Tagu deposit is highly fractionated and peraluminous and it has the tectonic signature of syn-collisional granites, which confirms that they were emplaced during the collision following the westward subduction of West Myanmar Terrane beneath Sibumasu during the Cretaceous to Tertiary periods [1,3,5,8,53].

### 4.3. Mechanism of Ore Formation

In the Tagu tin–tungsten deposit, the two-phase liquid-rich aqueous inclusions, two-phase vapor-rich fluid inclusions, a three-phases $H_2O$–$CO_2$–NaCl inclusions are the main fluid inclusion types. $CO_2$-bearing inclusions coexisting with vapor-rich and liquid-rich two-phase aqueous inclusions in groups are generally interpreted as evidence for fluid immiscibility [37]. [54] mentioned that tin–tungsten deposits in Myanmar were formed at less than 360 °C. In comparison, the homogenization temperature of the Tagu tin–tungsten deposit ranges from 140 °C to 405 °C. Furthermore, the fluid inclusion assemblage of the Tagu tin–tungsten deposit is different from those of other well-known world class tin–tungsten deposits in Myanmar, such as Mawchi and Hermyingyi [55,56]. [4] also indicated that homogenization temperatures range from 175 °C to 340 °C for the Mawchi vein system, in which liquid-rich two-phase liquid and vapor primary fluid inclusions are dominant, and $H_2O$–$CO_2$ fluid inclusions are absent. [54] also reported a detailed fluid inclusion study of the vein quartz and fluorite from the Hermyingyi deposit and other deposits. [54] showed that the majority of inclusions homogenized at about 250 °C, with some filling temperatures of vapor-rich inclusions up to 360 °C. Table 3 shows the correlation of mineralogical assemblages, deposit types, and fluid inclusion systems between the Tagu tin–tungsten deposit and other well-known tin–tungsten deposits in Myanmar, such as Mawchi, Hermyingyi and Yadanabon.

According to previous fluid inclusion studies, no $H_2O$–$CO_2$ fluid inclusions were found in granite-related tin–tungsten deposits in Myanmar except the granite-related tin–tungsten deposit of the Tagu area. Figure 14 shows typical fluid evolutions. It is considered that the parent and original fluid at Tagu was probably $CO_2$-bearing fluid which evolved to two-phase vapor-rich fluid and two-phase liquid-rich aqueous fluid by fluid immiscibility. The ore-forming fluid of the Tagu tin–tungsten deposit most likely encountered fluid immiscibility process during the early stage, which was characterized by similar homogenization temperatures and different salinities (Figure 14). The salinities of the vapor-rich inclusions are higher than those of the $CO_2$-rich inclusions, which may have resulted from $CO_2$ separation from the fluid as the temperature and pressure decreased. The escape of gases can lead to an increase in the salinity of the residual fluid [57]. Eventually, the later stage ore fluid was produced by the mixing of meteoric water circulated through adjacent metasedimentary rocks.

**Table 3.** Comparison of most important W–Sn deposits in southern Myanmar.

| No. | Deposit | Location (N, E) | Deposit Type | Terrane/Fold Belt | Host Rocks/(Ages) | Intrusions/(Ages) | Fluid System | Ore Mineralogy | References |
|---|---|---|---|---|---|---|---|---|---|
| 1. | Mawchi (Sn–W) (mine) | 18°45′, 97°10′ | Vein-type | Mogok–Mandalay–Mergui Metamorphic Belt | Metasediments of MawchiFormation equivalent to Mergui Group | Quartz veins and stockworks in both tourmalinized biotite granite (LA-MC-ICP-MS zircon 42–45 Ma) | $H_2O$–NaCl | Cassiterite, wolframite, scheelite, pyrite, chalcopyrite, arsenopyrite, molybdenite, bismuthinite, sphalerite, galena | [4,11,21] |
| 2. | Hermyingyi (Sn–W) (mine) | 14°15′, 98°35′ | Vein-type | Southern Mogok–Mandalay–Mergui Metamorphic Belt | Metasedimentary rocks of Mergui Group | Megacrystic biotite granite(U–Pb) SHRI MP Zircon age of 61.7 ± 1.3 Ma; LA-ICP-MS zircon ages of 70.5 ± 0.8 Ma and 68.9 ± 1.8 Ma) | $H_2O$–NaCl | Wolframite, cassiterite, molybdenite, pyrite, sphalerite, chalcopyrite, galena, bismuth, bismuthinite | [11] |
| 3. | Yadanabon (Sn–W) (mine) | 11°17′05″, 99°17′ | Vein-type, alluvial | Southern Mogok–Mandalay–Mergui Metamorphic Belt | Metasedimentary rocks of Mergui Group | Coarse-grained biotite Granite (U–Pb zircon age of 50.3 ± 0.6 Ma) | No data | Wolframite, cassiterite, molybdenite, bismuth, pyrite, bismuthinite, chalcopyrite | [3,11] |
| 4 | Tagu (Sn–W) (mine) | 12°14′05″, 98°59′54″ | Vein-type | Southern Mogok–Mandalay–Mergui Metamorphic Belt | Metasedimentary rocks of Mergui Group | Pophyritic biotite granite | $H_2O$–$CO_2$–NaCl | Cassiterite, wolframite, pyrite, chalcopyrite, arsenopyrite,pyrrhotite, molybdenite, sphalerite, native bismuth, galena | This study |

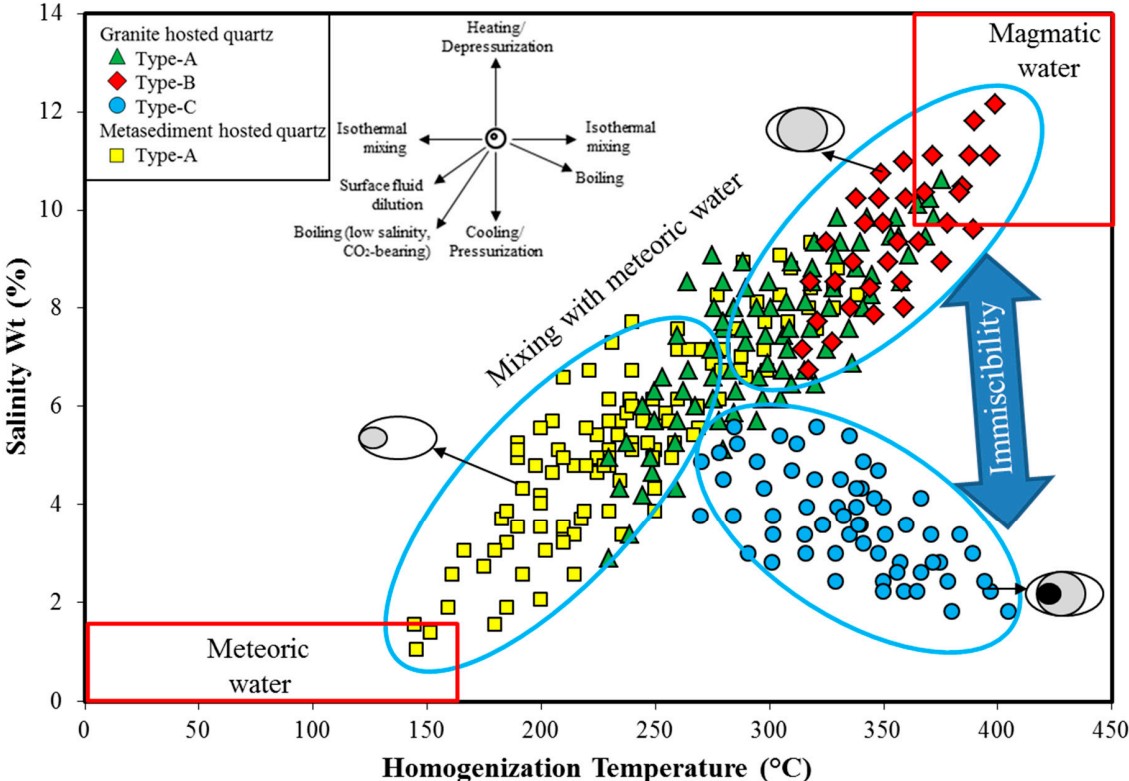

**Figure 14.** Plot of homogenization temperature (Th) versus salinity of fluid inclusion data from the Tagu Sn–W deposit with typical trends of fluid evolution patterns. [58].

## 5. Conclusions

The main conclusions from this study are summarized as follows:

(1) In the Tagu tin–tungsten deposit, nearly E–W trending vertical or steeply dipping mineralized quartz veins are hosted by both Cretaceous to Eocene granite and Carboniferous to Early Permian metasedimentary rocks. The granitic rocks are composed of quartz, feldspars (plagioclase, orthoclase, and microcline), and micas (muscovite and biotite). They are S-type and peraluminous granite, formed in a syn-collisional setting. The granites at Tagu show enrichment of LILEs, such as Rb, and K, and exhibit distinct negative anomalies of HFSEs, such as Nb, P, and Ti, indicating derivation of magma from the lower continental crust. The S-type granites at Tagu were produced through partial melting of the metasedimentary rocks;

(2) Early formed minerals are characterized by cassiterite and wolframite, which was followed by sulfide minerals. Deposition of these two major oxide ore minerals may have overlapped, but wolframite appears to be somewhat later than cassiterite. The fluid inclusions are characterized by three types of fluid inclusions: are two-phase vapor-rich fluid inclusions, two-phase liquid-rich fluid inclusions, and three-phase $H_2O$–$CO_2$–NaCl fluid inclusions. The parent mineralized fluid was probably $CO_2$-bearing fluid which evolved to two-phase vapor-rich fluid by fluid immiscibility. The ore-forming fluid of the Tagu tin–tungsten deposit encountered a fluid immiscibility process during its early stage, characterized by the similar homogenization temperature and different salinities. They have similar homogenization temperatures and are most likely derived from magmatic to post-magmatic hydrothermal fluids. The salinities of the vapor-rich fluid are higher than those of the $CO_2$-rich fluid, which may have resulted from $CO_2$ separation from the fluid as the temperature and pressure declined. The escape of gases can lead to an increase in the salinity of the residual fluid. Subsequently, ore fluid that was produced by magmatic water was probably mixed with meteoric water in the later stage, which may have circulated through the adjacent metasedimentary rocks. The present study strongly suggests that the ore-forming mechanisms of the Tagu tin–tungsten mineralization

are characterized by fluid immiscibility during an early stage and fluid mixing with meteoric water in the subsequent stage at a lower temperature. On the basis of fluid inclusion data, we suggested that tin–tungsten may have formed carbonate and bicarbonate complexes in $CO_2$-rich ore fluids, which supports the transport of tungsten as well as tin by these complexes during the formation of the Tagu tin–tungsten deposit.

**Author Contributions:** K.T.H., K.Y., T.T. and A.Z.M. carried out the fieldworks and developed the concepts, designed on this research. K.T.H. collected the data and samples and conducted the laboratory analysis and wrote this manuscript with contribution on discussion from K.Y. and K.W. All authors were contributed in reading, comments and giving the annotations on this manuscript.

**Funding:** This study was supported by Japan International Cooperation Agency (JICA), Shigen No Kizuna Program, and Japan.

**Acknowledgments:** We would like to express our gratitude to the Japan International Cooperation Agency (JICA), Shigen No Kizuna Program, and Japan for the PhD scholarship and their financial support. We are deeply grateful to Akira Imai, Department of Earth Resource Engineering, Kyushu University, Japan for his valuable annotations, suggestions and crucial reading repeatedly to advance our manuscript. We also express our appreciation to Khin Zaw, CODES Centre of Ore Deposits and Earth Sciences, University of Tasmania, Australia for his valuable suggestions, advice and critical reading which had substantially improved the earlier several versions of the manuscript. The authors are indebted to two journal reviewers and David Lentz for their constructive and valuable comments. Special thanks are also due to Diamond Shark Co. Ltd for generously making it possible to work in the mine site smoothly.

**Conflicts of Interest:** The authors declare no conflict of interest.

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
