# Peer review of "Petrogenesis, Ore Mineralogy, and Fluid Inclusion Studies of the Tagu Sn–W Deposit, Myeik, Southern Myanmar"

_minerals, doi:10.3390/min9110654_

Round 1

Reviewer 1 Report

Comments on minerals-624794 entitled ‘Petrogenesis, ore mineralogy and fluid inclusion studies of granite related Sn-W deposit at the Tagu area, Myeik Region, Southern Myanmar’ coauthored by Htun et al., submitted to Minerals

Htun et al. present a new dataset of petrography, lithogeochemistry, and fluid inclusions on the Tagu Sn-W deposit, Myeik Region, southern Myanmar, suggesting that ore-related biotite granite is strongly peraluminous S-type derived from partial melting of sedimentary rocks in a continental collisional setting, ore-fluids originating from the granite intrusions plus late mixing with meteoric water. The paper is well written and organized, and has important implications for metallogenic studies of Sn-W deposits genetically related to crustal-derived S-type granites. A minor revision is required to address the followings.

Tectonic setting of the Tagu granite intrusion needs more discussion, including use of some key references (e.g., Sylvester, 1989; Yang et al., 2018; for details, see comments on edited PDF manuscript) to support the arguments.

The information about the thickness of the Mergui Group is required, which would provide a clue as to why the emplacement of the granite took place in a thickened crustal section due to plate collision. Is the granite syn- or post-collisional (see Sylvester, 1989)?

What are errors in temperature measurements in fluid inclusion (FI) study? Such information is essential to a study of microthermometry, requiring to add to section 3.4.1. The V/(L+V) ratios of type-A FIs need be given, and do the same for type-B FIs.

Figure 1: See comment on this figure marked on the edited PDF manuscript.

Many minors are picked up for the authors’ consideration when a revision is made in order to improve the presentation of this paper.

Author Response

Dear Reviewer

Thank you very much for your constructive suggestions and review reports on our manuscript submitted to Minerals, titled ‘Petrogenesis, ore mineralogy and fluid inclusion studies of granite related Sn-W deposit at the Tagu area, Myeik Region, Southern Myanmar’. Based on the your comments, we have made the required modifications on this original manuscript. We provide below a point-by-point response to the comments.

Sincerely

Kyaw Thu Htun

Response to Reviewer 1 Comments

Point 1: Htun et al. present a new dataset of petrography, lithogeochemistry, and fluid inclusions on the Tagu Sn-W deposit, Myeik Region, southern Myanmar, suggesting that ore-related biotite granite is strongly peraluminous S-type derived from partial melting of sedimentary rocks in a continental collisional setting, ore-fluids originating from the granite intrusions plus late mixing with meteoric water. The paper is well written and organized, and has important implications for metallogenic studies of Sn-W deposits genetically related to crustal-derived S-type granites. A minor revision is required to address the followings.

Response 1:  Thank you very much for your constructive comments.

Point 2: Tectonic setting of the Tagu granite intrusion needs more discussion, including use of some key references (e.g., Sylvester, 1989; Yang et al., 2018; for details, see comments on edited PDF manuscript) to support the arguments.

Response 2:  Yes, we already included Yang et al., 2018 as you mentioned. (Line: 394,395).

Point 3: The information about the thickness of the Mergui Group is required, which would provide a clue as to why the emplacement of the granite took place in a thickened crustal section due to plate collision. Is the granite syn- or post-collisional (see Sylvester, 1989)?

Response 3:  Many thanks for your constructive comments and suggestions. The Mergui Group is a 3,000 m thick sequence of metasedimentary and sedimentary rocks comprising argillite and terrigenous clastic rocks (Brown & Heron 1923). (Line: 105, 106). The Tagu granite falls in the syn-collisional granites. (see Fig. 7B)

Point 4: What are errors in temperature measurements in fluid inclusion (FI) study? Such information is essential to a study of microthermometry, requiring to add to section 3.4.1. The V/(L+V) ratios of type-A FIs need be given, and do the same for type-B FIs.

Response 4:   The errors for the temperature measurements are described at (Line: 322,323). Already added to section 3.4.1. The V/(L+V) ratio is 0.15-0.35 for type A (Line: 342,343). The V/(L+V) ratio is 0.3-0.8 for type B (Line: 348,349).

Point 5: Figure 1: See comment on this figure marked on the edited PDF manuscript.

Response 5:   Yes, we already changed in figure 1.

Point 6: Line 14: Remove “lies” and change “as” to “is” put “a”

Response 6:    Okay, done. (Line: 14)

Point 7: Line 16: Remove “and”

Response 7:   Yes, already removed. (Line: 16)

Point 8: Line 18: Put “the”

Response 8:   Yes, done. (Line: 18)

Point 9: Line 22: Put “s”

Response 9:   Ok, Thanks! I did. (Line: 22)

Point 10: Line 23: Put “;” and “s”

Response 10:   Okay, added. (Line: 23)

Point 11: Line 24: Change “CO2 – liquid + CO2 – vapor” to “CO2-liquid + CO2- vapor”?

Response 11:   Yes, I changed. (Line: 24)

Point 12: Line 28, 29, 32 and 34: Correct “wt.  %” to wt.%

Response 12:  Yes, already corrected. (Line: 28,30, 33 and 35)

Point 13: Line 65, 66: Remove “in WGP” and “in the WGP” and put “,”.

Response 13:  Yes, I did. (Line: 67- 69)

Point 14: Line 77: Change “Despite” to “Although”.

Response 14:   Okay, already changed. (Line: 79)

Point 15: Line 84: Remove this legend down to the lower left corner and Make the arrow smaller, i.e. the size 'N' is comparable to 'LEGEND'.

Response 15:   Ok, legend on the map has already removed to the lower left corner of the Map in Figure 1. North line was deleted because north index has already included on the map. (Line: 86)

Point 16: Line 92: put “from” and remove “and these”

Response 16:   Yes, I corrected. (Line: 94)

Point 17: Line 93: Put “that”

Response 17:   Yes, done. (Line: 95)

Point 18: Line 94: Remove “, found”

Response 18:  Okay, I did. (Line: 96)

Point 19: Line 95: Remove “are found to”

Response 19:   Okay, already removed. (Line: 97)

Point 20: Line 97 and 98: Move this to line 101 just after '[9,19-21]'. How thick is this group?

Response 20:   Yes, already moved. (Line: 103-105). The thickness is expressed at (Line: 105, 106)

Point 21: Line 105: Remove “-SSE”

Response 21: Okay, I did. (Line: 110)

Point 22: Line 115: Remove “-SW”

Response 22: Yes, done. (Line: 120)

Point 23: Line 119: Remove “-SSE”

Response 23: Okay, I did. (Line: 125)

Point 24: Line 128: Remove “-SSE”

Response 24: Yes, done. (Line: 134)

Point 25: Line 132: Reference?

Response 25: Actually, I drew this map myself based on field data. I removed “modified after Tagu mine project map”. (Line: 138)

Point 26: Line 137 and 138: Remove “ed” put “s” and remove “by”. Rewording “The Mergui Pluton is elongated NNW-SSE parallel to the metasedimentary rocks of Mergui Group.”

Response 26: Okay, I reworded as “The elongate granite plutons emplaced along the NNW-SSE, the regional strike of the Mergui Group”. (Line: 143-145)

Point 27: Line 141: Change “in only” to “only in”

Response 27: Okay, done. (Line: 148)

Point 28: Line 149: Remove “metasedimentary rock” and “;”

Response 28: Yes, I did. (Line: 156)

Point 29: Line 162: Change “Photographs” to “Photomicrographs of granitoids and greywacke at” and remove “showing microscopic thin sections”

Response 29: Thank you for your comments. I changed. (Line: 170)

Point 30: Line 167: Remove “photomicrograph of metasedimentary rocks (Greywacke).

Response 30: Yes, I did. (Line: 175,176)

Point 31: Line 169: Change “wt %” to “wt.%” and replace “including” to “(in ppm) and”

Response 31: Okay, done. (Line: 211)

Point 32: Line 171: Remove “the highlighted” in table 1.

Response 32: Yes, already removed. (Line: 213)

Point 33: Line 181 and 182: Change “(20t)” to (20 t)”, “(30kV)” to “(30 kv)” and “70mA” to “(70 mA)”.

Response 33: Okay, I did. (Line: 186-188)

Point 34: Line 183 and 184: Change “Inductively Coupled Plasma-Mass Spectrometry” to “inductively coupled plasma-mass spectrometry”

Response 34: Yes, done. (Line: 189,190)

Point 35: Line 187,188,191 and 194: Change “2h” to “2 h”, “1h” to “1 h”, “1h” to “1 h” and “2h” to “2 h” and remove “accurate”.

Response 35:  Okay, already changed. (Line: 193, 194, 197 and 200)

Point 36: Line 197 and 198: Change “wt %” to “wt.%”.

Response 36:  Already changed. (Line: 226,227)

Point 37: Line 200 and 201: Remove “Line 200” and change “wt. %” to “wt.%”.

Response 37: Okay, I did. (Line: 203,204)

Point 38: Line 215: Remove “and”

Response 38: Already removed. (Line: 221)

Point 39: Line 217: This is problematic, requiring to rephrase.

Response 39:  I agree, I also think this is problematic. That’s why, I removed this sentence and Fig. 6E as well. (Line: 223,224)

Point 40: Line 225: Change “Wt  %” to “wt.%”

Response 40: Yes, I changed. (Line; 231,234,235)

Point 41: Line 235: Explain these circled in the figure caption, WPG- within-plate granite, etc.

Response 41:  Yes, I explained all circled in the Fig. 7B caption, such as syn-COLG- syn-collision granite, WPG- within-plate granite, VAG- volcanic-arc granite and ORG- ocean ridge granite. (Line: 248, 249)

Point 42: Line 247: Check the caption in Figure 8.

Response 42:  Yes, I already checked and changed the caption in Figure 8 such as Figure 8. Chondrite normalized REE diagram (A) and primitive mantle normalized trace element diagram (B) for Tagu granitoids. (Line: 259,260)

Point 43: Line 261: Remove “-SWW”

Response 43:  Okay, already removed. (Line: 274)

Point 44: Line 263: Remove “of study” and replace “bearing” to “hosted”

Response 44: Yes, done. (Line: 276, 278)

Point 45: Line 268: Change “as” to “by”

Response 45:  Okay, already changed. (Line: 281)

Point 46: Line 274: Change “of” to “within”

Response 46:  Yes, done. (Line: 287)

Point 47: Line 296: Change “contemporaneous” to “contemporaneously”

Response 47:  Yes, done. (Line: 306)

Point 48: Line 321: Secondary?

Response 48:  The fluid inclusions are clusters, isolation within the crystal growth zones are considered as primary. The fluid inclusions which is found at healing fracture are recognized as secondary fluid inclusions.

Point 49: Line 325: What is the ratios of V/(L+V)?

Response 49:  V/(L+V) is 0.15-0.35 for type A. (Line: 342,343)

Point 50: Line 331: What is the ratios of V/(L+V)?

Response 50:  V/(L+V) is 0.3-0.8 for type B. (Line: 348,349)

Point 51: Line 353: Change “wt. %” to “wt.%”.

Response 51:  Yes, done. (Line: 369)

Point 52: Line 360: Change “wt%NaCl eq.” to “wt.% NaCleqv.” in Table 2.

Response 52:  Yes, done. (Line: 376)

Point 53: Line 364: Change “Wt (%)NaCl equiv.” to “(wt.% NaCleqv.)” in figure 13B,D and F.

Response 53:  Okay, already changed. (Line: 380)

Point 54: Line 366: put “in”

Response 54:  Yes, done. (Line: 382)

Point 55: Line 372: change “wt. %” to “wt.%”

Response 55: Yes, done. (Line: 388)

Point 56: Line 378: Remove “extremely” and change “crust” to “collisional zones, even in Neoarchean time (e.g., Yang et al., 2018)”

Response 56: Yes, I changed. (Line: 394-395)

Point 57: Line 420: put “s”

Response 57:  Yes, done. (Line: 441)

Point 58: Line 423: remove “an”

Response 58: Okay, done. (Line: 444)

Reviewer 2 Report

since this study only have major and trace element data of whole rock, and FI data. you should focus on interpretation the mineralization processes in this deposit. your granite data can't solve too much problems. P vs. SiO2 could be used to identify I type granites from S type granites. some other trace element plot might also be helpful. Please read more Lithos papers to get some ideas for granite classification. 

Author Response

Dear Reviewer

Thank you very much for your constructive suggestions and review reports on our manuscript submitted to Minerals, titled ‘Petrogenesis, ore mineralogy and fluid inclusion studies of granite related Sn-W deposit at the Tagu area, Myeik Region, Southern Myanmar’. Based on your comments, we have made the required modifications on this original manuscript. We provide below a point-by-point response to the comments. 

Sincerely

Kyaw Thu Htun

Response to Reviewer 2 Comments

Point 1: Since this study only have major and trace element data of whole rock, and FI data. You should focus on interpretation the mineralization processes in this deposit. Your granite data can't solve too much problems. P vs. SiO2 could be used to identify I type granites from S type granites. Some other trace element plot might also be helpful. Please read more Lithos papers to get some ideas for granite classification.

Response 1:  Many thanks for your constructive comments and suggestions which make us improving in thinking and considerations about my research.

Point 2: Line 2,3 and 4: Put “,” and remove “granite related Sn-W deposit at” in Title and replace “deposit” to “area”

Response 2:   Thank you for your suggestions. Actually, we just want to mention about granite related Sn-W deposit to be clear for the readers when they read the title, they can know easily about what is writing in this paper. But, I follow your suggestion. I changed them.

Now the title is changed to “Petrogenesis, ore mineralogy and fluid inclusion studies of the Tagu Sn-W deposit, Myeik Region, Southern Myanmar” as suggested by reviewer.

Point 3: Line 17: check the definition of LILEs and HFSEs.

Response 3:   Yes, we already checked.

Point 4: Line 18: How do you identify this is the parental magma of Tagu deposits. field evidence? age?

Response 4:   Tagu granite itself is greisenized and hosts mineralized veins. Thus, it is clear that the ore fluid responsible for Sn-W mineralization was derived from the magmatic hydrothermal system which produces granitic magma emplacement and successive hydrothermal veining. The genetic implication of Tagu granite is defined on both field evidence and geochemical data.

Point 5: Line 18: how about upper crustal sedimentary rocks, which S-type granite is derived from.

Response 5:   Gechemical characteristics such as primitive mantle normalized trace element diagram suggesting that Tagu granite derived from the lower continental crust by differentiation.

Point 6: Line 22: evolution sequence, which one is earlier and which one is later, you have to identify it from mineral assemblage and crosscutting relationships, totally confusing readers in the following discussion.

Response 6:   Thank you for your constructive suggestions and comments. In this study, quartz in the vein hosted in granite are corresponding with earlier in which type A, type B and type C included, and quartz in the vein hosted in metasedimentary rocks are corresponding with later in which only type A included. (line: 25-26).

Point 7: Line 47: what are you trying to say? “It has one of the most diverse natural and mineral resources in Southeast Asia, largely reflecting its geological history”

Response 7:   Okay, we changed this sentence a little bit to be reasonable. (Line: 48)

Point 8: Line 55: make it clear, readers can't find this from map directly. “Southeast Asia Tin Belt”

Response 8:   We changed “Southeast Asia Granitoid Belts”. (Line: 56)

Point 9: Line 107: workings?

Response 9:   Yes, we changed as occurrences. (Line: 112)

Point 10: Line 113: Change “sometimes” to “locally”

Response 10:   Yes, done. (Line: 119)

Point 11: Line 137: wrong.

Response 11:   Yes, we changed such as “which are hosts to mineralized quartz veins.” (Line: 143)

Point 12: Line 200: Remove “line: 200”

Response 12:   Okay, already removed. (line: 203)

Point 13: Line 244: Lu depletion is not always discussed.

Response 13:   We are sorry about that. In this paper, we focus on other trace elements and REE for geochemical interpretation.

Point 14: Line 377: how could you know this is S type? high peraluminous? sys collision setting? garnet in granite? based on what? usually author use one paragraph to discuss this, you can't just say its I-type or S-type without any solid evidence.

Response 14:   S-type granite is defined by generally high peraluminous and syn-collisional setting.

Point 15: Line 386: according to major and trace element to discuss magma source seems impossible. if you want to discuss this, you have to show solid isotopic data. otherwise these discussion is meaningless.

Response 15:   Thank you for your constructive suggestions. I agree with you. We have a plan to conduct the isotopic data in future as further study. Although solid isotopic data are needed to show, we think, recent data are reasonable and sufficient to assert this interpretation.

Point 16: Line 400: MMM Belt, can’t use abbreviation unless you mentioned above.

Response 16:   Okay, we changed in the paragraph such as “Mogok-Mandalay-Mergui Belt (MMM) Belt”. (Line: 417)

Point 17: Line 404: Remove line “404” and “405”

Response 17:  Thank you for your constructive suggestions. But we want to retain to refer the [47] as they also noted similar collisional setting from the metallogeny of the Central Andean Tin Belt. (Line: 421,422) Ref; [47] changed to [48]

Point 18: Line 459: Remove “Pb”

Response 18:   Okay, done. (Line: 480)
